# Intervening Effects and Molecular Mechanism of Quercitrin on PCV2-Induced Histone Acetylation, Oxidative Stress and Inflammatory Response in 3D4/2 Cells

**DOI:** 10.3390/antiox11050941

**Published:** 2022-05-11

**Authors:** Qi Chen, Yuheng Wei, Yi Zhao, Xiaodong Xie, Na Kuang, Yingyi Wei, Meiling Yu, Tingjun Hu

**Affiliations:** College of Animal Science and Technology, Guangxi University, Nanning 530005, China; chenqi@st.gxu.edu.cn (Q.C.); weiyuheng@st.gxu.edu.cn (Y.W.); zhaoyi@st.gxu.edu.cn (Y.Z.); 1718304007@st.gxu.edu.cn (X.X.); 987204855@st.gxu.edu.cn (N.K.); weiyingyi@gxu.edu.cn (Y.W.); yumeiling@gxu.edu.cn (M.Y.)

**Keywords:** quercitrin, PCV2, histone acetylation, oxidative stress, inflammation

## Abstract

Porcine circovirus type 2 (PCV2) is the main pathogen causing porcine circovirus-associated diseases (PCVD/PCVADs), and infection of the host induces immunosuppression. Since quercitrin (QUE) has anti-inflammatory and antiviral activity, it is worth exploiting in animal diseases. In this study, the interventional effects and the molecular mechanism of QUE on PCV2-induced oxidative stress and inflammatory responses in 3D4/2 cells and the modulation of histone acetylation modifications were investigated. The ROS production was measured by DCFH-DA fluorescent probes. HAT and HDAC enzyme activity were determined by ELISA. Histone acetylation, oxidative stress and inflammation-related gene expression levels were measured by q-PCR. Histone H3 and H4 (AcH3 and AcH4) acetylation, oxidative stress and inflammation-related protein expression levels were measured by Western blot. The results showed that QUE treatment at different concentrations on PCV2-infected 3D4/2 cells was able to attenuate the production of ROS. Moreover, QUE treatment could also intervene in oxidative stress and decrease the enzyme activity of HAT and the mRNA expression level of HAT1, while it increased the enzyme activity of HDAC and HDAC1 mRNA expression levels and downregulated histone H3 and H4 (AcH3 and AcH4) acetylation modification levels. In addition, QUE treatment even downregulated the mRNA expression levels of IL-6, IL-8, IκB, AKT and p38, but upregulated the mRNA expression levels of IL-10, SOD, GPx1, p65, Keap1, Nrf2, HO-1 and NQO1. As to protein expression, QUE treatment downregulated the levels of iNOS, p-p65 and IL-8 as well as the phosphorylation expression of IκB and p38, while it upregulated the levels of HO-1 and NQO1. It was shown that QUE at 25, 50 or 100 μmol/L regulated p38MAPK and PI3K/AKT signaling pathways by downregulating cellular histone acetylation modification levels while inhibiting the NF-κB inflammatory signaling pathway and activating the Nrf2/HO-1 antioxidant signaling pathway, thus regulating the production of inflammatory and antioxidant factors and exerting both anti-inflammatory and antioxidant effects.

## 1. Introduction

Porcine circovirus type 2 (PCV2) is an icosahedrally symmetrical, non-capsular DNA virus containing a covalently closed single-stranded circular negative strand [1]. PCV2 infection can cause immunosuppression, and it often occurs in mixed infections with other pathogenic microorganisms in clinic and is the cause of postweaning multisystemic wasting syndrome (PMWS) [2], porcine dermatitis and nephropathy syndrome (PDNS), proliferative and necrotizing pneumonia (PNP), porcine respiratory disease syndrome (PRDC), reproductive failure (RD), congenital tremor (CT) [3] and other related diseases in newborn piglets. Therefore, it has caused huge economic losses to major pig breeding countries worldwide.

Macrophages are an important line of defense of the body’s immune system. They are involved in the development and progression of inflammatory diseases by regulating the secretion of cytokines, chemokines and other inflammatory mediators. In addition, histone modifications are major regulators of macrophage function and directly affect the body’s regulation of immune and inflammatory responses [4]. Furthermore, disease development is usually accompanied with the interaction of oxidative stress and inflammation. It was found that viral infection promotes the secretion of cytokines and chemokines and activates a pro-inflammatory signaling cascade to exacerbate the inflammatory response [5]. Related studies have reported that viral infection of immune cells induces oxidative stress and inflammatory responses, leading to the accumulation of ROS in cells. ROS further aggravate cellular oxidative damage by regulating the level of cellular histone acetylation and affecting the transcription levels of immune cell-related genes [6]. PCV2 infection of immune cells causes cells to produce large amounts of ROS, which induces the onset of oxidative stress in cells. PCV2 infection enhances the sensitivity of host cells to oxidative factors, resulting in a state of oxidative stress. Oxidative stress can mediate histone acetylation modifications in cells which, in turn, can mediate the expression of inflammation-related genes, thereby exacerbating the disease process [7].

Flavonoids are natural compounds that are widely found in plants and have pharmacological effects such as antibacterial, antiviral and antitumor. In addition, flavonoids have the ability to reduce or scavenge free radicals and modulate body immunity. Moreover, flavonoids play an anti-inflammatory role by inhibiting the activity of inflammatory immune cells, suppressing the production and expression of inflammatory genes and mediators and by regulating inflammation-related signaling pathways [8,9]. In addition, the flavonoid monomeric compound, QUE, could also recover pulmonary epithelial cell hyperplasia, and inflammatory cell infiltration improved via the tNF-κB/COX-2 and PGE2 signaling pathway [10].

Flavonoids exert antioxidant effects by scavenging excessive ROS production from the body. BinMowyna [11] showed that kaempferol, by activating the SIRT1 signaling pathway, upregulating the enzymatic activities of SOD and GSH, upregulating the protein expression of Bcl-2, downregulating the expression of CYP2E1 and PARP-1 and reducing the acetylation level of FOXO-1, then inhibited cellular oxidative stress and apoptosis and played a protective role against liver damage caused by acetaminophen (APAP) in animals. In addition, Malus doumeri leaf flavonoids (MDLFs) could increase the activity and gene expression levels of CAT, SOD, GSH and GSH-Px; downregulate the gene expression levels of MAPK, NF-κB and TNF-α; inhibit H_2_O_2_-induced oxidative stress in HEK293T cells, exerting antioxidant, anti-apoptotic and anti-inflammatory effects [12].

Flavonoids have a wide range of biological activities such as anti-inflammatory, antioxidant and immunomodulatory [13]. QUE belongs to the natural flavonoid monomer compounds and, thus, has the same biological activity. In addition, the development of animal diseases is often associated with inflammation or viral infections. Therefore, it is worthwhile to develop and exploit QUE for its role in the field of animal diseases.

Based on a previously established model of inflammation and oxidative stress in PCV2-infected 3D4/2 cells [14], we aimed to elucidate the intervening effects and molecular mechanism of QUE on PCV2-induced histone acetylation, oxidative stress and inflammatory response in 3D4/2 cells and to provide a theoretical basis for the exploitation of QUE and the prevention and treatment of PCVD/PCVADs.

## 2. Materials and Methods

### 2.1. Reagents

Quercitrin (standard product, HPLC purity ≥ 98%, product number: SQ8050, penicillin mixture (100×)) was purchased from Solarbio (Beijing, China). DMSO was purchased from Solarbio (Beijing, China). DMEM and fetal bovine serum were purchased from Gibco (Grand Island, NY, USA). Fetal bovine serum (FBS) and trypsin EDTA solution were purchased from BI (Shanghai, China). RNAiso Plus was purchased from TaKaRa (Dalian, China); fluorescent quantitative from Genstar (Beijing, China). CCK8 and HiScript III RT SuperMix (R312-01) were purchased from Novozymes (Nanjing, China); the ROS kit and BCA protein concentration assay kit were purchased from Biyuntian Biotechnology (Shanghai, China). Acetylcysteine (NAC) was purchased from (Shanghai, China). The Ultrasensitive ECL chemiluminescence kit (P10300) was purchased from Xin Saimei Biotechnology (Suzhou, China). AcH3 (#9649), AcH4 (#8647), NF-kB p65 (#6956), Phospho-NF-kB p65 (#8214S), p38 MAPK (#8690), Phospho-p38 MAPK (#8203S), IκB-α (#8219S), Phospho-IκB-α (#8219S), AKT (#4691T), Phospho-AKT (#9271T), Anti-mouse IgG (#7076P2), Anti-rabbit IgG (#7074S), β-actin (#4970) and HRP secondary antibodies (#7076, 7074) were purchased from CST (Mass, USA). IL-8 was purchased from Proteintech (Wuhan, China). iNOS was purchased from NOVUS (MN, USA). HO-1 (ab189491) and NQO1 (ab80588) were purchased from Abcam (Cambridge, UK).

### 2.2. Viruses and Cells

The porcine circovirus II (PCV2) strain was gifted by the Laboratory of Animal Disease Diagnosis and Immunity, Nanjing Agricultural University. The viral solution was obtained by the laboratory by amplification culture on PK-15 cells, and its viral titer TCID_50_ was 10^−5^/0.1 mL, measured according to the Reed–Muench method [15]. Porcine alveolar macrophage cell line 3D4/2 cells were obtained from the cell bank of Wuhan University.

### 2.3. Methods

#### 2.3.1. The Determination of 3D4/2 Cell Viability with CCK8

##### The Determination of QUE on the Viability of 3D4/2 Cells with CCK8

The experiment was set-up with a control group, a 0.05% DMSO group (containing 0.05% DMSO) and QUE groups with seven concentrations of quercitrin (i.e., 6.25, 12.5, 25, 50, 100, 200 and 400 μmol/L, diluted with a 2.5% FBS–DMEM culture medium) as shown in Table 1. The concentration of the 3D4/2 cells was adjusted to 1 × 10^5^ cells/mL and seeded in 96-well cell culture plates with a 100 μL/well cell suspension. All groups were incubated in a cell incubator at 37 °C with 5% CO_2_ for 36 h. Then, at 1 h before the end of the incubation, the culture medium was discarded and washed 3 times with PBS. Afterwards, 100.0 µL of serum-free DMEM medium containing 10.0 µL of CCK8 was added to each well, and the cells were continued to be cultured for 1.5 h at 37 °C with 5% CO_2_.

##### The Determination of QUE on the Viability of PCV2-Infected 3D4/2 Cells with CCK8 Method

The experimental grouping is shown in Table 2. The control group was incubated by adding 100 µL of serum-free DMEM culture medium per well, while the PCV2-infected groups, 0.05% DMSO + PCV2 group or QUE groups with six concentrations were incubated by adding 100 µL of PCV2 virus solution (MOI = 1.0). All the well plates were adsorbed in an incubator at 37 °C with 5% CO_2_ for 2 h and then the PCV2 was removed. After washing the cells with PBS three times, 200 μL of 2.5% FBS–DMEM was added to each well for the control group and CCK8 blank control groups. To the QUE groups, 200 μL of QUE was added per well to the six concentrations. After incubation for another 36 h, the culture medium was discarded, and the plates were washed three times with PBS. Moreover, 100 µL of serum-free DMEM medium containing 10.0 µL of CCK8 was added to each well and incubated for another 1 h. The 96-well plate was taken out from the incubator using a light-proof bag, and the OD value was measured at 450 nm. Then, the maximum safe concentration was calculated according to the formula. The cell survival rate calculation formula was as follows: cell survival rate = (OD of test group-OD of blank control group)/(OD of control group-OD of blank control group) × 100%.

#### 2.3.2. DCFH-DA Detection for ROS Levels in 3D4/2 Cells

The experimental grouping is shown in Table 3. The cell concentration was adjusted to 2 × 10^5^ cells/mL, and cells were plated at 1.0 mL/well in 24-well plates. Afterwards, the cells were incubated at 37 °C and 5% CO_2_ in an incubator for 12 and 24 h, respectively. Afterwards, the DCFH-DA was diluted with serum-free DMEM culture medium at 1:1000, and 300 μL of DCFH-DA probe was added to each well of the 24-well plate and incubated for 20 min protected from light. The cells were washed 3 times with serum-free DMEM to remove the fluorescent probes that did not enter the cells. Then, 1.0 mL of PBS was added to each well, scraped off the cells and the cell concentration was adjusted to 1 × 10^5^ cells/mL. Next, the cells were seeded at 200 μL per well in a 96-well black plate to detect the value of OD, with the excitation wavelength at 488 nm and the emission wavelength at 525 nm.

#### 2.3.3. ELISA Detection for the Activity of Acetylase in 3D4/2 Cells 

The experimental grouping is shown in Table 4. The cell concentration was adjusted to 5 × 10^5^ cells/mL, and the cells were inoculated in 12-well plates at 1 mL/well and incubated overnight at 37 °C in 5% CO_2_. Afterwards, the supernatant was discarded, and the cells were washed 3 times with PBS. The PCV2-infected group and QUE groups were added with PCV2 (MOI = 1.0), while the control group was added with an equal amount of serum-free DMEM. All the plates were incubated for 2 h at 37 °C with 5% CO_2_, and then the PCV2 was removed. After washing the cells three times with PBS, 1.0 mL of 2.5% FBS–DMEM was added to each well for the cell control and PCV2-infected groups. Three concentrations of QUE (i.e., 100, 50 or 25 μmol/L) diluted by 2.5% FBS–DMEM were added to the QUE groups. Cells were incubated at 37 °C and 5% CO_2_ in an incubator for 8, 12, 24 or 36 h. Then, the samples were collected and repeatedly frozen–thawed three times. Afterwards, the cell supernatant was collected after centrifugation at 3000 rpm/min for 20 min and stored at −80 °C. The enzymatic activities of HAT and HDAC in the cells were determined according to the ELISA kit instructions. 

#### 2.3.4. q-PCR Detection for the Expression Levels of mRNA in 3D4/2

The experimental grouping was the same as that in Section 2.3.3. All cells were incubated for 8, 12, 24 or 36 h for subsequent detection. Briefly, total RNA extraction was performed on the collected samples. After RNA reverse transcription to cDNA, the transcription levels of HAT1, HDAC1, oxidative stress and inflammatory-related genes in the 3D4/2 cells were detected by q-PCR, and the expression of the target genes was calculated using 2^−^^ΔΔct^ with β-actin as the reference gene. The gene sequences are shown in the Appendix A (Appendix A).

#### 2.3.5. Western Blot Detection for the Expression Levels of Proteins in 3D4/2

The experiment grouping was the same as that in Section 2.3.3. All cells were incubated for 24 h for subsequent detection. Specifically, total proteins were first performed on the collected samples. Then, the sample protein concentration was determined according to the BCA kit’s instructions. Afterwards, the samples were subjected to SDS-PAGE electrophoresis and transferred to PVDF membranes using the constant current method for 1.5 h. Furthermore, the PVDF membranes were closed overnight at 4 °C using 5% skim milk. The next day, PVDF membranes were washed three times with 1 × TBST. Subsequently, the diluted primary antibody solution was incubated with PVDF membranes at 4 °C overnight. Similarly, PVDF membranes were washed three times with 1 × TBST. The PVDF membranes were placed in the diluted secondary antibody solution and incubated in an incubator at 37 °C for 1 h. Finally, ultrasensitive ECL chemiluminescence development was performed. The grayscale values of the protein bands were analyzed using ImageQuant TL analysis software, and the relative grayscale values of the target proteins were compared using GAPDH/β-actin as the internal reference values.

### 2.4. Data Processing and Analysis

One-way ANOVA (one-way analysis of variance) was used to analyze the test data, which were expressed as the mean ± SD. And all the data were statistically analyzed by the IBM SPSS Statistics 23.0 (Chicago, IL, USA). The shoulder label * indicates a significant difference compared to the control group (*p* < 0.05), and ** indicates a very significant difference compared to the control group (*p* < 0.01). In addition, the shoulder label # indicates a significant difference compared to the PCV2-infected group (*p* < 0.05), and ## indicates a very significant difference compared to the PCV2-infected group (*p* < 0.01).

## 3. Results

### 3.1. Effect of QUE on the Proliferative Activity of PCV2-Infected 3D4/2 Cells

As displayed in Figure 1A, QUE from 25 to 400 μmol/L significantly promoted the activity of 3D4/2 cells, and this promotion was independent of DMSO. As shown in Figure 1B, after infecting PCV2, cell viability decreased significantly compared with the control group (*p* < 0.01). The addition of QUE at 12.5, 25, 50, 100, 200 or 400 μmol/L promoted the activity of the PCV2-infected cells (*p* < 0.01). In addition, the activity of PCV2-infected 3D4/2 cells was elevated with QUE treatment. Moreover, 0.05% DMSO had no significant effect on the activity of PCV2-infected cells.

### 3.2. Effect of Quercitrin on ROS Levels in PCV2-Infected 3D4/2 Cells In Vitro and the Regulation of Histone Acetylation Modification

As shown in Figure 2A, when compared with the control group, PCV2 infection of 3D4/2 cells for 12 or 24 h significantly elevated the production level of ROS, while NAC significantly decreased the production level of ROS (*p* < 0.01). QUE at 25, 50 or 100 μmol/L was able to attenuate ROS production at 12 or 24 h post-treatment when compared with the PCV2 group (*p* < 0.01).

As shown in Figure 2C,E, HAT activity significantly increased at 8, 12, 24 or 36 h in PCV2-infected cells, and HDAC activity was extremely reduced at 8, 12 or 24 h post-infection when compared with the control group, but cellular HDAC activity significantly increased at 36 h post-infection (*p* < 0.05 or *p* < 0.01). Compared with the PCV2-infected group, QUE at 25 or 50 μmol/L significantly decreased HAT activity at 8, 12, 24 or 36 h post-treatment (*p* < 0.05 or *p* < 0.01), and QUE at 100 μmol/L significantly decreased HAT activity at 8, 24 or 36 h post-treatment (*p* < 0.05 or *p* < 0.01). In addition, QUE at 25 μmol/L significantly increased HDAC activity at 8 h post-treatment but significantly decreased HDAC activity at 36 h post-treatment in PCV2-infected 3D4/2 cells (*p* < 0.01). QUE at 50 or 100 μmol/L significantly increased HDAC activity at 24 h post-treatment but decreased HDAC activity at 36 h (*p* < 0.05 or *p* < 0.01) post-treatment. QUE at 100 μmol/L significantly increased HDAC activity at 12 or 24 h post-treatment (*p* < 0.05).

As shown in Figure 2B,D, compared with the control group, PCV2 infection significantly upregulated the expression levels of HAT1 mRNA and significantly downregulated the expression levels of HDAC1 mRNA in 3D4/2 cells at 8, 12, 24 or 36 h post-infection (*p* < 0.05 or *p* < 0.01). Compared with the PCV2-infected group, QUE treatment at 25 μmol/L for 8, 12 or 36 h significantly downregulated HAT1 mRNA expression levels in PCV2-infected cells (*p* < 0.05 or *p* < 0.01), and QUE treatment at 50 or 100 μmol/L for 8, 12, 24 or 36 h significantly downregulated HAT1 mRNA expression levels in PCV2-infected cells (*p* < 0.05 or *p* < 0.01). The activity of HDAC had no significance post-QUE treatment at 25 μmol/L for 8, 12, 24 or 36 h. QUE treatment at 50 μmol/L for 12 or 24 h significantly promoted the mRNA expression levels of HDAC1 in PCV2-infected cells (*p* < 0.05 or *p* < 0.01), QUE treatment at 100 μmol/L for 8, 12 or 24 h significantly promoted the mRNA expression levels of HDAC1. These findings indicate that QUE could inhibit the expression level of HAT1 mRNA in PCV2-infected 3D4/2 cells and promote the mRNA expression level of HDAC (see also Figure 2D,E).

As shown in Figure 2F,G, PCV2 infection of 3D4/2 cells for 24 h significantly upregulated the acetylation modification levels of histones AcH3 and AcH4 compared with the control group (*p* < 0.01). Compared with the PCV2-infected group, QUE at 25, 50 or 100 μmol/L was able to attenuate the acetylation levels of histones H3 or H4 at 24 h post-treatment (*p* < 0.01). This indicates that QUE could regulate the levels of histone acetylation modification in PCV2-infected 3D4/2 cells.

### 3.3. Effect of QUE on the Expression of Genes Related to the Induction of Oxidative Stress and Inflammatory Response in PCV2-Infected 3D4/2 Cells

As shown in Figure 3, PCV2 infection in 3D4/2 cells was able to upregulate the expression levels of cellular IL-6 and IL-8 mRNA at 8, 12, 24 or 36 h compared with the control group (*p* < 0.05 or *p* < 0.01). In addition, p65 mRNA expression levels decreased in PCV2-infected 3D4/2 cells at 8, 12 or 24 h post-infection (*p* < 0.05 or *p* < 0.01), and IL-10 mRNA expression levels decreased in PCV2-infected 3D4/2 cells at 8 and 12 h post-infection (*p* < 0.05). Compared with the PCV2-infected group, QUE from 25 to 100 μmol/L significantly downregulated the mRNA expression levels of IL-6 and IL-8 at 8 or 36 h post-treatment (*p* < 0.05 or *p* < 0.01) (see also Figure 3A,B) and significantly upregulated mRNA expression of p65 at 12 or 24 h post-treatment (*p* < 0.01 or *p* < 0.05) (Figure 3C). QUE at 100 μmol/L significantly upregulated p65 mRNA gene expression at 8, 12 or 24 h post-treatment (*p* < 0.05) (Figure 3C). QUE at 25, 50 or 100 μmol/L extremely upregulated mRNA expression levels of IL-10 (*p* < 0.01) at 12 or 24 h post-treatment. In addition, 50 and 100 μmol/L of QUE extremely upregulated IL-10 mRNA expression at 8 or 36 h post-treatment (*p* < 0.01) (Figure 3D).

As shown in Figure 4A,B, 3D4/2 cells infected with PCV2 upregulated the intracellular IkB and AKT mRNA expression levels. However, QUE was able to downregulate the intracellular IkB and AKT mRNA expression levels post-QUE treatment. Specifically, compared with the PCV2 group, QUE at 25, 50 or 100 μmol/L was able to extremely downregulate the mRNA expression levels of IκB at 8 and 12 h post-QUE treatment (*p* < 0.01 or *p* < 0.05). Interestingly, three concentrations of QUE treated for 36 h significantly upregulated the mRNA expression levels of IκB in comparison with the PCV2 group (*p* < 0.01 or *p* < 0.05). Similarly, QUE treatment at 100 μmol/L was able to significantly downregulate the mRNA expression levels of AKT at 8, 12 or 24 h post-QUE treatment (*p* < 0.01 or *p* < 0.05).

Furthermore, as shown in Figure 4C,D, 3D4/2 cells infected with PCV2 downregulated the intracellular mRNA expression levels of SOD and Gpx1. However, QUE was able to upregulate the mRNA expression levels of SOD and Gpx1. Specifically, QUE at 100 μmol/L was able to extremely upregulate the mRNA expression levels of SOD post-treatment at 8, 24 or 36 h in comparison with the PCV2 group (*p* < 0.01 or *p* < 0.05). Similarly, QUE at 100 μmol/L was able to extremely upregulate the mRNA expression levels of Gpx1 post-treatment at 12 or 24 h (*p* < 0.01 or *p* < 0.05).

As shown in Figure 5A, 3D4/2 cells infected with PCV2 upregulated the mRNA expression levels of p38. However, QUE was able to downregulate the mRNA expression level of p38. Specifically, compared with the PCV2 group, QUE at 25, 50 or 100 μmol/L was able to extremely downregulate the mRNA expression levels of p38 post-treatment at 12 or 36 h (*p* < 0.05 or *p* < 0.01).

Furthermore, as shown in Figure 5B–E, 3D4/2 cells infected with PCV2 downregulated the mRNA expression levels of Keap1, HO-1, Nrf2 or NQO1. However, QUE was able to upregulate the mRNA expression levels of Keap1, HO-1, Nrf2 and NQO1 mRNA expression levels. Specifically, QUE at 25, 50 or 100 μmol/L was able to extremely upregulate the mRNA expression levels of Keap1 or HO-1 post-treatment at 12 h (*p* < 0.05 or *p* < 0.01). Similarly, QUE at 25, 50 or 100 μmol/L was able to extremely upregulate the mRNA expression levels of HO-1 or NQO1 post-treatment at 24 h in comparison with the PCV2 group (*p* < 0.05 or *p* < 0.01).

### 3.4. Effect of QUE on Signaling Pathways Related to Oxidative Stress and Inflammatory Response in PCV2-Infected 3D4/2 Cells

From Figure 6A–D, PCV2-infected 3D4/2 cells at 24 h significantly upregulated the proteins expression levels of iNOS and IL-8 compared with the control group (*p* < 0.05). Compared with the PCV2-infected group, QUE could differentially downregulate the protein expression levels of iNOS and IL-8 post-treatment at 24 h. QUE treatment at 50 or 100 μmol/L could attenuate the expression protein levels of iNOS (*p* < 0.05), and QUE at 25 μmol/L also showed a tendency to downregulate the protein expression level of IL-8 in PCV2-infected cells (*p* < 0.05). 

As shown in Figure 7, compared with the control group, PCV2 infection of 3D4/2 cells at 24 h significantly increased the levels of total p65 protein phosphorylation and IκB protein phosphorylation (*p* < 0.05). Compared with the PCV2-infected group, QUE treatment from 50 to 100 μmol/L at 24 h was able to downregulate the protein expression level of p-p65 in PCV2-infected 3D4/2 cells, and QUE treatment from 25 to 100 μmol/L extremely downregulated the protein phosphorylation levels of IκB (*p* < 0.01). These results indicate that PCV2-infected 3D4/2 cells could upregulate the expression levels of total cellular protein p65 and IκBα phosphorylation, while the phosphorylation of p65 and nucleation were inhibited post-QUE treatment.

As shown in Figure 8, compared with the control group, the protein expression levels of HO-1 and NQO1 were significantly attenuated in PCV2-infected 3D4/2 cells (*p* < 0.05 or *p* < 0.01). Compared with the PCV2-infected group, QUE treatment at 50 or 100 μmol/L could significantly inhibit the reduction of HO-1 protein expression in PCV2-induced 3D4/2 cells, and QUE at 25 or 100 μmol/L could significantly inhibit the protein expression levels of NQO1 in PCV2-infected cells post-treatment at 24 h (*p* < 0.05).

As shown in Figure 9, PCV2 infection of 3D4/2 cells significantly upregulated the protein phosphorylation expression levels of cellular p38 and AKT compared with the control group (*p* < 0.05). Compared with the PCV2-infected group, QUE treatments from 25 to 100 μmol/L could significantly reduce the phosphorylation level of p38 post-treatment at 24 h (*p* < 0.05 or *p* < 0.01). In addition, QUE treatment at 50 or 100 μmol/L significantly attenuated the phosphorylation levels of cellular AKT (*p* < 0.05).

## 4. Discussion

Natural herbal medicines have certain regulatory effects on cellular oxidative stress, acetylation and inflammatory responses. ROS are a series of key signaling molecules in the oxidative stress response and play an important role in the regulation of the physiological and pathological processes of the body. If cells are subjected to severe harmful stimuli, the balance between the body’s production of ROS and the elimination of antioxidant systems becomes imbalanced, ROS production continues and mitochondria are damaged, which will then lead to oxidative damage and inflammatory responses in cells. Sun [7] found that PCV2 infection of PK-15 cells promoted intracellular ROS production and triggered the translocation of HMGB1 from the nucleus to the cytoplasm, which releases the nuclear HMGB1’s restriction of PCV2 replication. The results of the present study are consistent with the above results, as PCV2 infection caused a significant increase in the level of intracellular ROS production in 3D4/2 cells, inducing the cells to be in a state of oxidative stress. Natural antioxidant substances of plant origin can effectively interfere with oxidative stress and reduce the production of harmful byproducts of ROS, which are important for inhibiting oxidative stress damage and preventing related diseases [16]. In this study, QUE treatments from 25 to 100 μmol/L were found to be effective in scavenging ROS in PCV2-infected 3D4/2 cells, indicating that QUE could interfere with PCV2-induced oxidative stress.

Histones are the main protein components of chromatin in eukaryotic cells. When histones are modified, they cause structural changes in chromatin and play a role in regulating eukaryotic genes. Histone modifications mainly include histone acetylation modification, phosphorylation modification, methylation modification, histone ubiquitination, SUMO modification and biotinylation modification [17]. Epigenetic regulators have been reported to promote viral replication and proliferation, with histone acetylation modifications playing an important role. Histone acetylation is jointly regulated by HAT and HDAC, and an organism is in a dynamic balance of acetylation and deacetylation under normal conditions. Viral infection leads to an imbalance in the level of acetylation modifications in host cells, and the imbalance would regulate the expression of inflammatory mediators that are often associated with the development of multiple diseases [18]. The results of this study showed that PCV2 infection of 3D4/2 cells significantly decreased HDAC enzyme activity and increased HAT enzyme activity. HDC1 and HAT1 mRNA expression levels were consistent with the trend in enzyme activity. This suggests that PCV2 infection caused changes in the levels of cellular HAT/HDAC and induced a state of cellular acetylation modification. Meanwhile, PCV2 infection of 3D4/2 cells could significantly elevate the acetylation levels of histone H3 and histone H4. Moreover, this process led to the production of numerous ROS in infected cells, which remains consistent with a previous study by Yang [6]. Natural herbs have modulating effects on cellular oxidative stress, acetylation and inflammatory responses. Yang also found that *Sophora subprosrate* polysaccharide could induce epigenetic modifications by increasing HDAC1 expression and downregulating acetylation modifications of Ac-H3 and Ac-H4, which could balance the histone acetylation modifications in PCV2-infected RAW267.4 cells. Moreover, *Sargassum* polysaccharide strongly inhibited PCV2-induced histone acetylation and cytokine production. Therefore, it can protect the host from damage and improve the resistance to PCV2 [19]. In addition, the flavonoid, luteolin, enhanced histone H3 acetylation and increased the expression of Fas and FasL through activation of the ERK and JNK pathways and, thus, was able to induce apoptosis in human leukemia HL-60 cells [20]. Likewise, *Acanthopanax senticosus* also promoted histone H3 acetylation through inhibition of histone deacetyltransferase activity which, in turn, induced FasL expression in HL60/ADM cells and induced apoptosis in a dose- and time-dependent manner in human leukemia cells [21]. In this experiment, QUE was found to differentially downregulate HATase activity and the mRNA expression level of HAT1 in PCV2-infected 3D4/2 cells. In addition, it could also upregulated HDACase activity and the mRNA expression level of HDAC1. Meanwhile, QUE at 25, 50 or 100 μmo/L also significantly downregulated the acetylated proteins expression levels of histones H3 and H4. This indicates that QUE could regulate the level of histone acetylation modification in PCV2-infected 3D4/2 cells, which is beneficial for 3D4/2 cells to fight against various damages caused by PCV2 infection and to maintain cellular homeostasis. Moreover, PCV2 infection induced cells to produce ROS transmission signals, which might cause changes in histone acetylation levels, thus acting as a genetic switch. QUE could downregulate the acetylation modification levels of histones AcH3 and AcH4 in 3D4/2 cells by regulating ROS levels in PCV2-infected 3D4/2 cells which, in turn, regulated the transcription of genes related to inflammatory mediators. These findings indicate that the acetylation modification was related to cellular oxidative stress and inflammatory response.

Cytokines and inflammatory mediators play an important role in the formation and release of inflammation. PCV2 is the main pathogen of PMWS, which usually infects porcine alveolar macrophages [22], attacking the lymphoid tissue of the body, leading to immunosuppression and, eventually, developing into immunodeficiency disease. In this experiment, 3D4/2 cells were used as a model of inflammation. The results of this experiment showed that infection of 3D4/2 cells with PCV2 at 8 or 36 h significantly upregulated the mRNA expression levels of inflammatory factors IL-6 and IL-8. Meanwhile, the mRNA expression levels of IL-10 showed a trend of decreasing and then increasing, which might have been caused by the complex role of IL-10 in acute viral infection. QUE could provide effective control of the cellular inflammatory response by effectively inhibiting the mRNA expression levels of IL-6 and IL-8 while promoting the mRNA expression levels of IL-10. This is consistent with the findings that QUE has in vitro anti-inflammatory activity [23,24,25].

Oxidative stress and inflammatory response interact during viral infection to induce immunosuppression in the body and trigger a series of diseases. The main mechanism is to promote the production of antioxidant enzymes SOD, HO-1, NQO1 and other non-enzymatic antioxidants, such as GSH and GPx1, through the binding of Nrf2 to target genes, thus improving the antioxidant capacity of the body and reducing the degree of oxidative damage. In addition, the body can regulate the expression of related pro-inflammatory factors through an immunomodulatory system centered on kinases such as NF-κB, AP-1 and MAP. It was found that RSV infection downregulated the mRNA expression level of Nrf2 in airway epithelial cells. The nuclear localization reduction of Nrf2 protein in infected cells induced rapid ROS production and was the main cause of lung inflammation and oxidative damage [5]. In this study, PCV2 infection caused a decrease in the mRNA expression levels of SOD, GPx1, Keap1, Nrf2, HO-1 and NQO1 in 3D4/2 cells. In addition, the mRNA expression levels of SOD, GPx1, Keap1, Nrf2, HO-1 and NQO1 in PCV2-infected cells was promoted to different degrees post-QUE treatment. This suggests that QUE can inhibit oxidative stress through the antioxidant defense system. Meanwhile, oxidative stress occurs in an organism causing the body to stimulate intracellular signaling cascades such as NF-κB to enhance the expression of pro-inflammatory factors [26]. NF-κB is one of the downstream target proteins of the PI3K/AKT signaling pathway. AKT is a downstream kinase of PI3K and, therefore, plays a key regulatory role in the NF-κB and MAPK signaling pathway. AKT activates IkB kinase (IKKα), which leads to IκB degradation, thereby allowing NF-κB to translocate into the nucleus [27]. Activated NF-κB promotes the expression of multiple genes to induce disease [28]. p38 Mitogen activates protein kinase (p38MAPK) and is closely associated with oxidative stress or inflammatory responses [29,30,31], playing a key role in signaling a cascade of responses to various cellular stimuli [32]. p38MAPK pathway activators are roughly the same as that of the JNK pathway and include inflammatory pro-inflammatory factors, oxidative stress, UV or heat shock stimuli. Activated p38MAPK further activates NF-κB, a key signaling molecule for early nuclear transcription factors which, in turn, regulates a range of inflammatory responses and releases cytokines and inflammatory mediators. In this study, the inflammation caused by PCV2 might be associated with activation of mitogen activation-related protein kinases (MAPKs). On the other hand, activation of the nuclear factor-κB protein family was associated with triggering the overproduction of pro-inflammatory factors and inflammatory mediators in the body. PCV2 infection of 3D4/2 cells promoted elevated mRNA expression levels of IκB, AKT and p38. However, PCV2 infection of 3D4/2 cells at 8 and 24 h suppressed the mRNA expression level of cellular p65, but the cellular mRNA expression levels of p65 picked up at 36 h post-infection. This result is inconsistent with the results of previous studies, which might be that viral infection of the organism upregulated the mRNA expression level of IκBα and downregulated the mRNA expression level of p65. In this way, the balance between NF-κB p65 and IκBα was disrupted, which led to the activation of the NF-κB signaling pathway. In this experiment, QUE was able to effectively inhibit the mRNA expression levels of related pathway genes, such as IκB, AKT and p38, and regulated cellular inflammation and oxidative damage [33]. This is consistent with the reported anti-inflammatory and oxidative effects of QUE. This suggests that the interventional effect of QUE on viral infection may be related to the mRNA expression levels of NF-κB, Nrf2/HO-1, PI3K/AKT and p38MAPK pathway genes. Therefore, this should be subsequently verified by detecting the protein levels of related signaling pathways.

In this study, PCV2 infection of 3D4/2 cells elevated the protein phosphorylation levels of cellular p65 and IκB, leading to an inflammatory response in the cells. This is consistent with the findings of Yang [20] that PCV2 infection of 3D4/2 cells promoted the secretion of TNF-α, IL-1β and IL-6 in 3D4/2 cells, activated the NF-κB signaling pathway and induced a cellular inflammatory response. Additionally, IL-8, also known as CXCL8, is an important member of the C-X-C chemokine family. In this study, IL-8 protein levels were significantly upregulated in infected 3D4/2 cells. This indicates that the chemokine IL-8 plays an important role in the inflammatory response induced by PCV2-infected porcine alveolar macrophages. In addition, related reports found that PCV2 infection upregulated the related genes including IL-8 and IL-1α [34,35], which is consistent with the results of this experiment. iNOS is a class of enzymes that use oxidative stress of nitric oxide to assist macrophages in fighting pathogens in the immune system. iNOS is one of the most important target genes mediated by NF-κB [36] and is closely associated with oxidative damage and inflammatory responses in an organism [37,38,39]. In this study, the protein expression level of iNOS significantly increased post-PCV2 infection, which might be related to the activation of NF-κB/iNOS signaling and induction of PCV2 inflammatory response in vitro.

The Keap1–Nrf2 signaling pathway is a classical antioxidant signaling pathway. The nuclear factor, Nrf2, is a core transcription factor that acts as a major regulator in the antioxidant response against exogenous stimuli and toxicants [40]. Common downstream gene products include NQO1, HO-1, GSTs and GCLC [41]. In addition, NQO1 and HO-1 are two important downstream regulatory genes of Nrf2 and play key roles in the expression of phase II antioxidant enzymes [42]. HO-1 regulates redox reactions and is the rate-limiting enzyme in heme catabolism. It catalyzes the production of biliverdin from heme, and biliverdin forms bilirubin, both of which are powerful free radical scavengers in the body and protective against oxidative stress-induced cellular damage [43]. Furthermore, NQO1 is a flavin protease present in the cytoplasm. As one of the important phase II antioxidant enzymes downstream of intracellular Nrf2, it catalyzes the single-step two-electron reduction reaction of quinones, preventing their participation in the redox cycle and the production of reactive oxygen species [44]. In this study, PCV2 infection of 3D4/2 cells induced downregulation of NQO1 and HO-1 protein expression which, in turn, caused oxidative damage and an inflammation response. In addition, the downregulation of NQO1 and HO-1 might be due to the upregulation of protein phosphorylation levels of p38, and it was also possible that the downregulation of these two proteins was due to the overproduction of ROS caused by PCV2 infection which, in turn, activated the NF-κB pathway and inhibited the Nrf2 pathway.

MAPKs, including extracellular signal-regulated kinases (ERK1/2), C-JUN, JNK/SAPK and mitogen-activated protein kinase p38MAPK, are commonly activated by inflammation-causing activation by cytokines [45]. Oxidative stress induces receptor-dependent apoptosis and disrupts mitochondria in normal cells. Mitochondrial dysfunction will further increase the accumulation of ROS and specifically activate ERKs, JNKs or the MAPK pathway [46,47,48]. Thus, ROS is one of the key factors in the activation of the MAPK signaling pathway. The PCV2-infected cells in this study activated the p38MAPK signaling pathway and upregulated the protein expression level of p-p38. This might be related to the excessive production of intracellular ROS caused by PCV2.

PI3K/AKT is a classical intracellular signaling pathway that activates the downstream effector serine/threonine kinase AKT by promoting phosphorylation of its Thr308 and Ser473 sites. AKT is a downstream kinase of PI3K and plays a key regulatory role in the NFκB and MAPK signaling pathways. Studies have reported that PCV2-infected 3D4/2 cells can upregulate AKT, p38MAPK, ERK and IκB protein phosphorylation expression levels by activating the PI3K/AKT and p38MAPK pathways and the NFκB signaling pathway [49]. In this experiment, AKT protein phosphorylation levels in infected cells were elevated post-PCV2 infection, suggesting that PCV2 might activate the AKT pathway. This is consistent with the results of the mentioned study.

## 5. Conclusions

In conclusion, QUE attenuated the phosphorylated expression levels of NF-κBp65, p38MAPK and AKT proteins; upregulated the protein expression levels of NQO1 and HO-1; downregulated the protein expression levels of IL-8 and iNOS in PCV2-infected 3D4/2 cells. It suggested that QUE might promote the activation of the Nrf2 signaling pathway by inhibiting the phosphorylated expression of p38MAPK and AKT proteins. Moreover, QUE further inhibited the activation of the NF-κB signaling pathway and suppressed the phosphorylation and degradation of IκBα in PCV2-infected 3D4/2 cells.

## Figures and Tables

**Figure 1 antioxidants-11-00941-f001:**
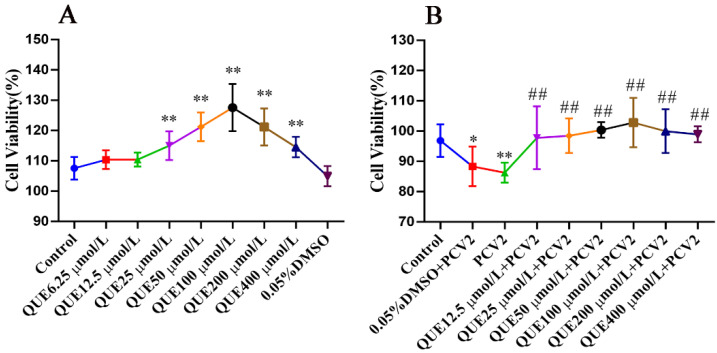
The viability of 3D4/2 cells (%, *n* = 6). (**A**) Control group; QUE6.25 μmol/L: 6.25 μmol/L QUE; QUE12.5 μmol/L: 12.5 μmol/L QUE; QUE25 μmol/L: 25 μmol/L QUE; QUE50 μmol/L: 50 μmol/L QUE; QUE100 μmol/L: 100 μmol/L QUE; QUE200 μmol/L: 200 μmol/L QUE; QUE400 μmol/L: 400 μmol/L QUE; 0.05% DMSO: culture medium containing 0.05% DMSO. ** *p* < 0.01. (**B**) Control: control group; 0.05% DMSO + PCV2: PCV2 + 0.05% DMSO; PCV2: PCV2-infected group; QUE12.5 μmol/L: PCV2 + 12.5 μmol/L QUE; QUE25 μmol/L: PCV2 + 25 μmol/L QUE; QUE50 μmol/L: PCV2 + 50 μmol/L QUE; QUE100 μmol/L: PCV2 + 100 μmol/L QUE; QUE200 μmol/L: PCV2 + 200 μmol/L QUE; QUE400 μmol/L: PCV2 + 400 μmol/L QUE. * *p* < 0.05, ** *p* < 0.01 and ## *p* < 0.01.

**Figure 2 antioxidants-11-00941-f002:**
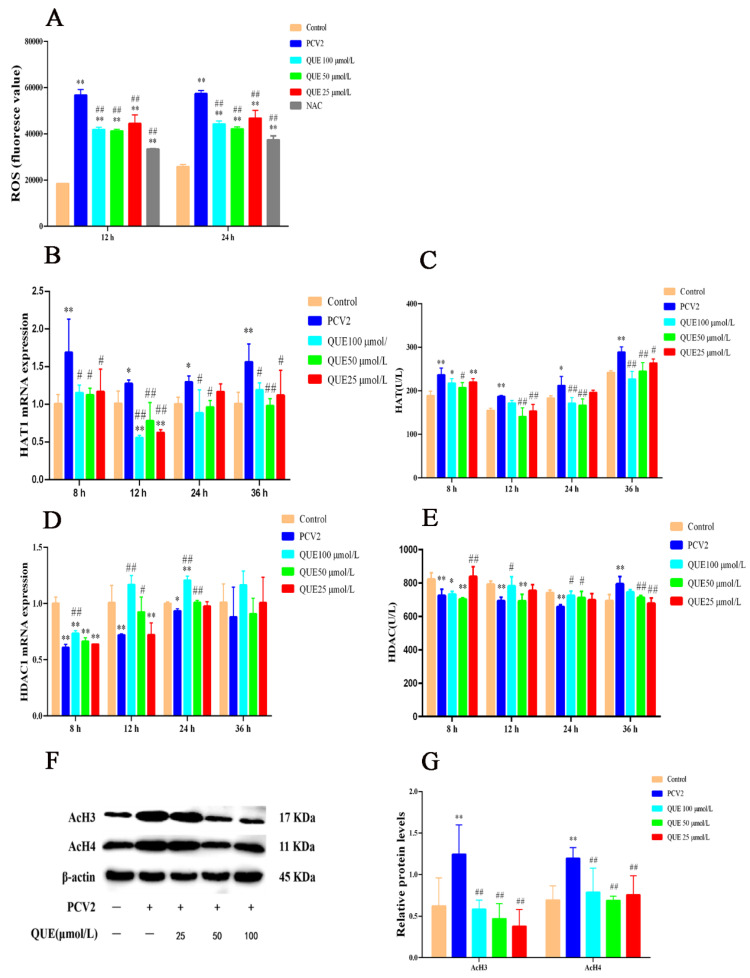
Effect of QUE on ROS levels and its regulation of histone acetylation modification in PCV2-infected 3D4/2 cells in vitro. (**A**) The production level of ROS. (**B**,**C**) The mRNA expression level and enzyme activity of HAT. (**D**,**E**) The mRNA expression level and enzyme activity of HDAC. (**F**,**G**) The expression levels of histones AcH3 and AcH4. Control: control group; PCV2: PCV2-infected group; QUE100 μmol/L: PCV2 + 100 μmol/L QUE; QUE50 μmol/L: PCV2 + 50 μmol/L QUE; QUE25 μmol/L: PCV2 + 25 μmol/L QUE; NAC: 1 mM NAC + PCV2. * *p* < 0.05, ** *p* < 0.01, # *p* < 0.05 and ## *p* < 0.01.

**Figure 3 antioxidants-11-00941-f003:**
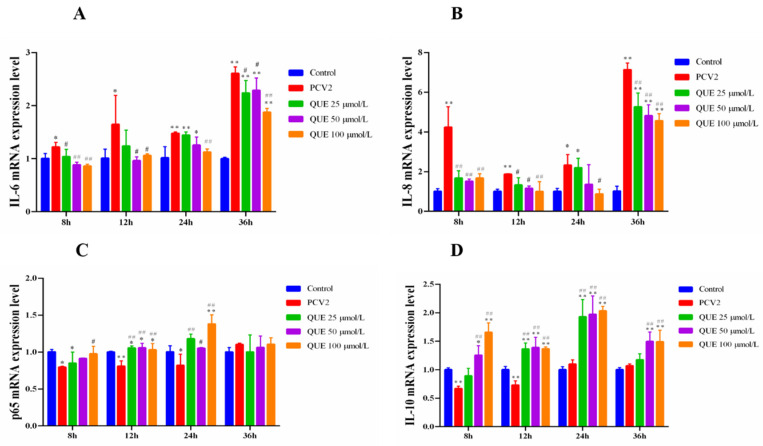
Effect of QUE on IL-6, IL-8, P65 and IL-10 mRNA expression of genes in PCV2-infected 3D4/2 cells. (**A**–**D**) The mRNA expression levels of IL-6, IL-8, p65 and IL-10. Control: control group; PCV2: PCV2-infected group; QUE100 μmol/L: PCV2 + 100 μmol/L QUE; QUE50 μmol/L: PCV2 + 50 μmol/L QUE; QUE25 μmol/L: PCV2 + 25 μmol/L QUE. * *p* < 0.05, ** *p* < 0.01, # *p* < 0.05 and ## *p* < 0.01.

**Figure 4 antioxidants-11-00941-f004:**
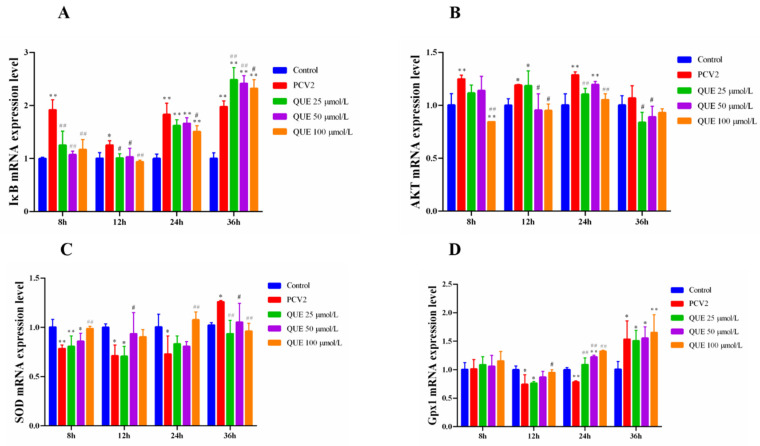
Effect of QUE on IκB, AKT, SOD and Gpx1 mRNA expression of genes in PCV2-infected 3D4/2 cells. (**A**–**D**) The mRNA expression levels of IκB, AKT, SOD and Gpx1. Control: control group; PCV2: PCV2-infected group; QUE100 μmol/L: PCV2 + 100 μmol/L QUE; QUE50 μmol/L: PCV2 + 50 μmol/L QUE; QUE25 μmol/L: PCV2 + 25 μmol/L QUE. * *p* < 0.05, ** *p* < 0.01, # *p* < 0.05 and ## *p* < 0.01.

**Figure 5 antioxidants-11-00941-f005:**
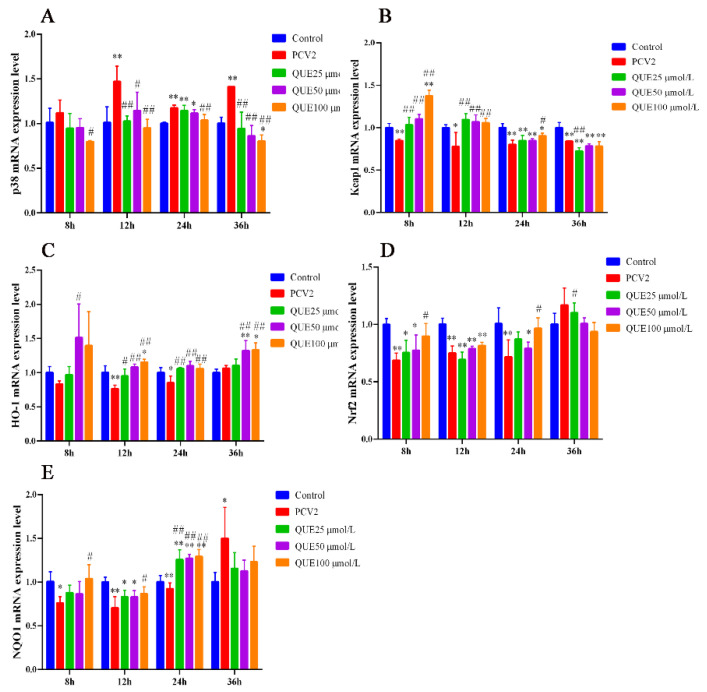
Effect of QUE on p38, Keap1, HO-1, Nrf2 and NQO-1 mRNA expression of genes in PCV2-infected 3D4/2 cells. (**A**–**E**) The mRNA expression levels of p38, Keap1, HO-1, Nrf2 and NQO1. Control: control group; PCV2: PCV2-infected group; QUE100 μmol/L: PCV2 + 100 μmol/L QUE; QUE50 μmol/L: PCV2 + 50 μmol/L QUE; QUE25 μmol/L: PCV2 + 25 μmol/L QUE. * *p* < 0.05, ** *p* < 0.01, # *p* < 0.05 and ## *p* < 0.01.

**Figure 6 antioxidants-11-00941-f006:**
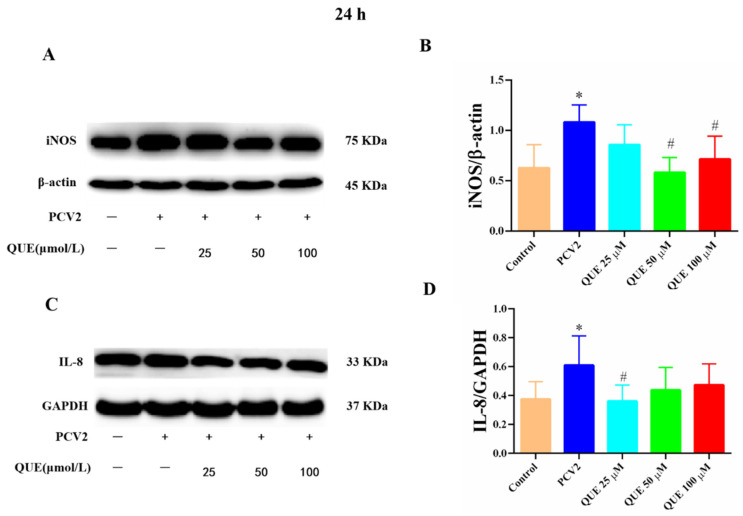
Effects of QUE on iNOS and IL-8 protein expression in PCV2-infected 3D4/2 cells. (**A**–**D**) The protein expression levels of iNOS and IL-8. Control: control group; PCV2: PCV2-infected group; QUE100 μmol/L: PCV2 + 100 μmol/L QUE; QUE50 μmol/L: PCV2 + 50 μmol/L QUE; QUE25 μmol/L: PCV2 + 25 μmol/L QUE. * *p* < 0.05 and # *p* < 0.05.

**Figure 7 antioxidants-11-00941-f007:**
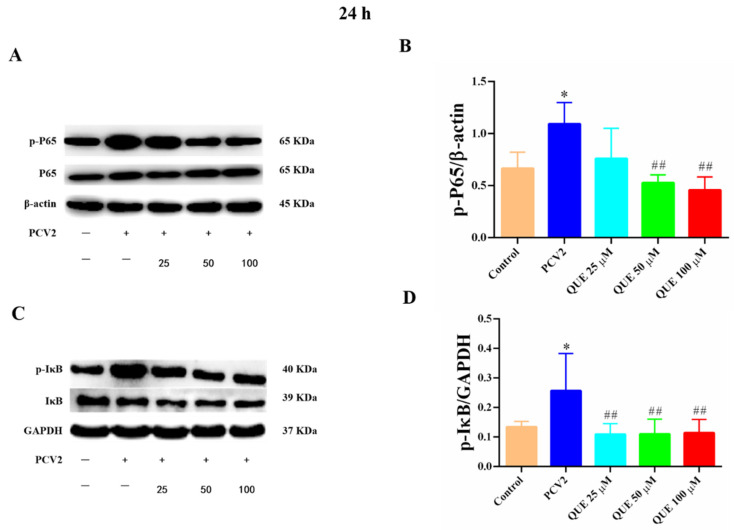
Effects of QUE on p-p65 and p-pIκB protein expression in PCV2-infected 3D4/2 cells. (**A**–**D**) The protein expression levels of p-P65 and p-IκB. Control: control group; PCV2: PCV2-infected group; QUE100 μmol/L: PCV2 + 100 μmol/L QUE; QUE50 μmol/L: PCV2 + 50 μmol/L QUE; QUE25 μmol/L: PCV2 + 25 μmol/L QUE. * *p* < 0.05 and ## *p* < 0.01.

**Figure 8 antioxidants-11-00941-f008:**
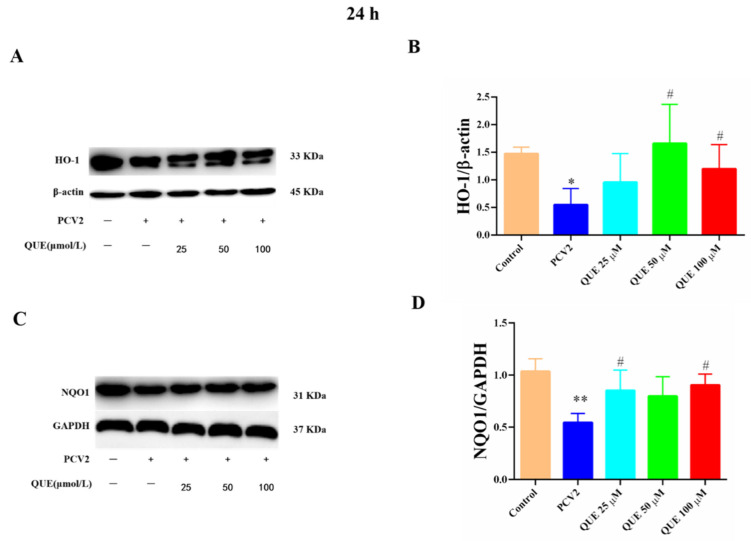
Effects of QUE on HO-1 and NQO-1 protein expression in PCV2-infected 3D4/2 cells. (**A**–**D**) The protein expression levels of HO-1 and NQO1. Control: control group; PCV2: PCV2-infected group; QUE100 μmol/L: PCV2 + 100 μmol/L QUE; QUE50 μmol/L: PCV2 + 50 μmol/L QUE; QUE25 μmol/L: PCV2 + 25 μmol/L QUE. * *p* < 0.05, ** *p* < 0.01 and # *p* < 0.05.

**Figure 9 antioxidants-11-00941-f009:**
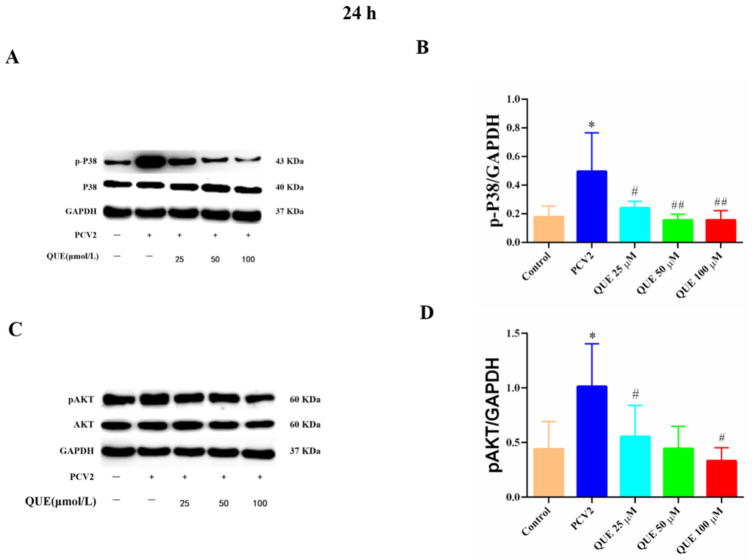
Effects of QUE on P38 and pAKT protein expression in PCV2-infected 3D4/2 cells. (**A**–**D**) The protein expression levels of p-P38 and pAKT. Control: control group; PCV2: PCV2-infected group; QUE100 μmol/L: PCV2 + 100 μmol/L QUE; QUE50 μmol/L: PCV2 + 50 μmol/L QUE; QUE25 μmol/L: PCV2 + 25 μmol/L QUE. * *p* < 0.05, # *p* < 0.05 and ## *p* < 0.01.

**Table 1 antioxidants-11-00941-t001:** Experimental groups of QUE on the proliferation of 3D4/2 cells.

Groups	Treatments
Control group	2.5% FBS–DMEM medium
0.05% DMSO group	Containing 0.05% DMSO medium
QUE 12.5 μmol/L	12.5 μmol/L QUE
QUE 25 μmol/L	25 μmol/L QUE
QUE 50 μmol/L	50 μmol/L QUE
QUE 100 μmol/L	100 μmol/L QUE
QUE 200 μmol/L	200 μmol/L QUE
QUE 400 μmol/L	400 μmol/L QUE

QUE, quercitrin; DMSO: dimethyl sulfoxide.

**Table 2 antioxidants-11-00941-t002:** Experimental grouping of QUE on PCV2-infected 3D4/2 cell viability.

Groups	Treatments of Virus	Treatments
Control group	Serum-free medium	2.5% FBS–DMEM medium
0.05% DMSO	PCV2 MOI = 1.0	Containing 0.05% DMSO medium
PCV2 group	PCV2 MOI = 1.0	2.5% FBS–DMEM medium
PCV2 + QUE 12.5 μmol/L	PCV2 MOI = 1.0	12.5 μmol/L QUE
PCV2 + QUE 25 μmol/L	PCV2 MOI = 1.0	25 μmol/L QUE
PCV2 + QUE 50 μmol/L	PCV2 MOI = 1.0	50 μmol/L QUE
PCV2 + QUE 100 μmol/L	PCV2 MOI = 1.0	100 μmol/L QUE
PCV2 + QUE 200 μmol/L	PCV2 MOI = 1.0	200 μmol/L QUE
PCV2 + QUE 400 μmol/L	PCV2 MOI = 1.0	200 μmol/L QUE

QUE, quercitrin; DMSO: dimethyl sulfoxide; PCV2, porcine circovirus type 2.

**Table 3 antioxidants-11-00941-t003:** Experimental grouping for ROS detection.

Groups	Treatments of Virus	Treatments
Control group	Serum-free medium	2.5% FBS–DMEM medium
PCV2 group	PCV2 MOI = 1.0	2.5% FBS–DMEM medium
PCV2 + QUE 25 μmol/L	PCV2 MOI = 1.0	25 μmol/L QUE
PCV2 + QUE 50 μmol/L	PCV2 MOI = 1.0	50 μmol/L QUE
PCV2 + QUE 100 μmol/L	PCV2 MOI = 1.0	100 μmol/L QUE
NAC 1 mmol/mL	PCV2 MOI = 1.0	1 mmol/mL NAC

QUE, quercitrin; PCV2, porcine circovirus type 2; NAC, acetylcysteine.

**Table 4 antioxidants-11-00941-t004:** Experimental grouping for acetylase level detection.

Groups	Treatments of Virus	Treatments
Control group	Serum-free medium	2.5% FBS–DMEM medium
PCV2 group	PCV2 MOI = 1	2.5% FBS–DMEM medium
PCV2 + QUE 25 μmol/L	PCV2 MOI = 1	25 μmol/L QUE
PCV2 + QUE 50 μmol/L	PCV2 MOI = 1	50 μmol/L QUE
PCV2 + QUE 100 μmol/L	PCV2 MOI = 1	100 μmol/L QUE

Note: QUE, quercitrin; PCV2, porcine circovirus type 2.

## Data Availability

Some or all data, models, or code generated or used in the study are available from the corresponding author by request. The data are not publicly available due to privacy.

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
