# Peer review of "Intervening Effects and Molecular Mechanism of Quercitrin on PCV2-Induced Histone Acetylation, Oxidative Stress and Inflammatory Response in 3D4/2 Cells"

_antioxidants, 2022, doi:10.3390/antiox11050941_

Round 1

Reviewer 1 Report

REF. REVIEW: antioxidants-1671903

RESEARCH PAPER: Intervening effects and molecular mechanism of Quercitrin on PCV2-induced histone acetylation, oxidative stress and inflammatory response in 3D4/2 cells

COMMENTS TO THE AUTHOR 

The authors examined the effect of different doses of quercitrin (QUE), a flavonoid derived from quercetin, on a macrophage-based cell culture challenged with porcine circovirus type 2 (PCV2). Although quercetin has been extensively tested for its antioxidant and anti-inflammatory properties, quercitrin does not have the same volume of publications presented by its precursor. Some studies have also demonstrated antioxidant and anti-inflammatory effects in some cellular models, as B16F10 melanoma cells and HepG2.  Basically, the experimental design involved 3 main groups of 3D4/2 cells (immortalized porcine alveolar macrophages): a) control 3D4/2; b) 3D4/2 + PCV2; and c) 3D4/2 + PCV2 + QUE; the latter has been divided into several treatment doses (from 12.5 to 400 umol/L), depending on the experiment to be performed. The parameters used to prove the protective intervention of QUE were basically related to oxidative stress and inflammation. Histone acetylation was valorized as PCV2 seems to induce inflammation by suppressing it on resident cells. The large number of results, according to the authors, show that quercitrin at 25 μmol/L, 50 μmol/L, 100 μmol/L regulated p38MAPK and PI3K/AKT signaling pathways by down-regulating cellular histone acetylation modification levels, while inhibiting NF-κB inflammatory signaling pathway and activating Nrf2/HO-1 antioxidant signaling pathway, thus regulating the production of inflammatory and antioxidant factors, and exerting both anti-inflammatory and antioxidant effects. The theme is relevant and results interesting. As mentioned, quercitrin does not show such results in the current literature as quercetin does. The experimental design is complex enough to fulfill its purpose. However, the reading is not fluid, and many of the elements necessary to understand the purpose of the study and the reason for having been constructed in this way need to be sought outside the text. The Introduction lacks crucial information, and the main aim is not clear. In the material and methods session many details and references are missing. Results show low quality graphs (too small) and discussion is extensive.

Major comments

  1. The introduction, in general, is very unspecific, repetitive and does not establish a linear line of reasoning. The importance of using an alveolar macrophage as a model is unclear, and the relationship between histones, inflammation and oxidative stress in the model used remains unclear. It is suggested to reconstruct with more specific and relevant information
  2. Materials and Methods section needs to be significantly improved before any type of publication is considered. Methods need further description. Few references are mentioned, and details are not presented. Some molecules detections are switched or missing.
  3. It is strongly recommended that the figures in the results section be enlarged, and the resolution improved. In some of them, in the printed version, it is impossible to see the subtitles. Legend patterns repeat and some graphics details need to be present.
  4. Discussion is too long and overexplained. Please, consider to unshorten.

Minor comments

  1. Page 2, lines 47-59: several experimental models are mixed, and a relation with citrus flavonoids suddenly appear. It is suggested to remove or redo.
  2. Page 2, lines 71-74: please, reference this paragraph.
  3. Page 2, lines 75-77: please, reference the previously established model
  4. Page 2, lines 77-81: Aim is not clear
  5. Page 3, lines 103-105: First time that the origin of 3D4/2 cells is mentioned
  6. Page 3, lines 107-123: What is the CCK8 method used for?
  7. Page 3, line 108: Please clarify why cells are inoculated, and not placed or plated, in culture plates
  8. Page 3, lines 124-132: No referenced method
  9. Page 3, line 132: Which enzyme marker?
  10. Page 3, line 133: How acetylase activity is modulated in the method described in 2.3.3.
  11. Page 3-4, lines138-154: Please, specify and reference which gene expression detection method is used here, why the expression of genes related to histone acetylation would reflect their activity, and why in the abstract the method of quantifying HAT and HDAC activity is described as ELISA.
  12. Page 4, lines 155-162: Authors say that they quantify iNOS, IL-8, NF-kB, HO-1, NQO1, P38, and PI3K/AKT by western blotting. However, in the results section, although WB is really showed, they present IL-6, IL-10, IL-8 and p65 mRNA expression levels. Where are the methods used to quantify these molecules expression? Some of them are even mentioned in Methods.
  13. Page 4, pages 163-169: Please explain here how replicates were constructed and used.
  14. Page 4, lines 171-177: Please clarify why you call CCK8 as a proliferative activity assay here. Still, it doesn't seem clear to me which viability decrease you would be dealing with: the one presented by the PCV2 group or the one shown by the 0.05% DMSO group
  15. Page 4, lines 177-179: This sentence is interpretative in a Results session.
  16. Page 5, lines 192-193: Why do authors present total ROS with HAT activity? How are they supposably related here?
  17. Page 5, lines 209 – 211: This sentence is interpretative in a Results session.
  18. Page 6, line 233: How ROS levels are regulating histone acetylation here? Please, clarify.
  19. Page 6, figures 2D – 2E: Please clarify if mRNA expression levels do not have a housekeeping normalization.
  20. Page 7, figures 3A – 3D: Please clarify if mRNA expression levels do not have a housekeeping normalization.
  21. Page 8, figures 4A – 4D: Please clarify if mRNA expression levels do not have a housekeeping normalization.
  22. Page 9, figure 5A – 5E: Please clarify if mRNA expression levels do not have a housekeeping normalization.
  23. Page 9, figure 5D: It does not seem very clear to me whether the expression of Nrf2 alone would be enough to prove its participation in the pathway, since its action is only effective upon translocation to the nucleus.

  1. Page 10, Discussion: There are many different routes of secondary AO produced by Nrf2/KEAP. Two important things should be considered here: a) How to differentiate the antioxidant from the anti-inflammatory effect, if they are linked and the results are similar? In addition, The AO effects involved in inflammation involve the Xanthine/XO system. This system was not evaluated and is related to the secondary response in Nrf2/KEAP system. Please, consider that.

Author Response

Reviewer #1:

Major comments:

  1. The introduction, in general, is very unspecific, repetitive and does not establish a linear line of reasoning. The importance of using an alveolar macrophage as a model is unclear, and the relationship between histones, inflammation and oxidative stress in the model used remains unclear. It is suggested to reconstruct with more specific and relevant information.

Response: Macrophages are an important line of defense of the body's immune system. They are involved in the development and progression of inflammatory diseases by regulating the production of cytokines, chemokines and other inflammatory mediators. In addition, histone modifications are major regulators of macrophage function and directly affect the body's regulation of immune and inflammatory responses [4]. Furthermore, disease development is usually accompanied by the interaction of oxidative stress and inflammation. It was found that viral infection promotes the secretion of cytokines and chemokines and activates a pro-inflammatory signaling cascade to exacerbate the inflammatory response [5]. Related studies have reported that viral infection of immune cells induces oxidative stress and inflammatory responses, leading to the accumulation of ROS in cells. ROS further aggravate cellular oxidative damage by regulating the level of cellular histone acetylation and affecting the transcription and expression of immune cell-related genes [6]. PCV2 infection of immune cells causes cells to secrete large amounts of ROS, which induces the onset of oxidative stress in cells. PCV2 infection enhances the sensitivity of host cells to oxidative factors, resulting in a state of oxidative stress. And oxidative stress can mediate histone acetylation modifications in cells, which in turn can mediate the expression of inflammation-related genes, thereby exacerbating the disease process [7].

  1. Materials and Methods section needs to be significantly improved before any type of publication is considered. Methods need further description. Few references are mentioned, and details are not presented. Some molecules detections are switched or missing.

Response: As required, in the section of materials and methods section, the following adjustments were made:

2.2: References were added as required.

2.3. Methods

2.3.1 The determination of 3D4/2 cell viability with CCK8

2.3.1.1 The determination of QUE on the viability of 3D4 / 2 cells with CCK8

The experiment was set up with cell control group, 0.05% DMSO group (containing 0.05% DMSO) and QUE groups with seven concentrations of quercitrin (6.25 μmol/L, 12.5 μmol/L, 25 μmol/L, 50 μmol/L, 100 μmol/L, 200 μmol/L and 400 μmol/L, diluted with 2.5%-FBS-DMEM culture medium) (As showed in Table 1). The concentration of the 3D4/2 cells was adjusted to 1×105 cells/mL and seeded in 96-well cell culture plates with 100 μL/well cell suspension. All the groups were incubated in a cell incubator at 37°C with 5% CO2 for 36 h. Then, at 1 h before the end of the incubation, the culture medium was discarded and gently washed 3 times with PBS. Afterwards, 100.0 µL of serum-free DMEM medium containing 10.0 µL of CCK8 was added to each well, and the cells were continued to be cultured for 1.5 h at 37°C with 5% CO2.

Table 1 Experimental grouping of QUE on proliferation of 3D4/2 cells

Groups

Treatments

Cell control group

2.5%-FBS-DMEM medium

0.05% DMSO group

Containing 0.05% DMSO medium

QUE 12.5 μmol/L

12.5 μmol/L QUE

QUE 25 μmol/L

25 μmol/L QUE

QUE 50 μmol/L

50 μmol/L QUE

QUE 100 μmol/L

100 μmol/L QUE

QUE 200 μmol/L

200 μmol/L QUE

QUE 400 μmol/L

400 μmol/L QUE

Note: QUE, quercitrin; DMSO: Dimethyl sulfoxide.

2.3.1.2 The determination of QUE on the viability of PCV2-infected 3D4 / 2 cells with CCK8 method

The experiment was set up with cell control group (As showed in Table 2). Cell control group was incubated by adding 100 µL of serum-free DMEM culture medium per well, while PCV2 infection group, 0.05% DMSO + PCV2 group or QUE groups with six concentrations were incubated by adding 100 µL of PCV2 virus solution (MOI=1.0). All the well plates were adsorbed in incubator at 37°C with 5% CO2 for 2 h and then the PCV2 solution was removed. After washing the cells with PBS three times, 200 μL of 2.5%-FBS-DMEM solution was added to each well for the cell control group and CCK8 blank control group, respectively. QUE group was added 200 μL of QUE solution in six concentrations per well respectively. After incubation for another 36 h, the culture medium was discarded and the plates were washed three times with PBS. 100 µL of serum-free DMEM medium containing 10.0 µL of CCK8 was added to each well and incubated for another 1 h. The 96-well plate was removed from the incubator using a light-proof bag and the OD value was measured at 450 nm on an enzyme marker. Then the maximum safe concentration was calculated according to the formula. Cell survival rate calculation formula: Cell survival rate = (OD of test group - OD of blank control group) / (OD of cell control group - OD of blank control group) × 100%.

Table 2 Experimental grouping of QUE on PCV2-infected 3D4/2 cells viability

Groups

Treatment of virus

Treatments

Cell control group

Serum-free medium

2.5%-FBS-DMEM medium

0.05% DMSO

PCV2 MOI=1

Containning 0.05% DMSO medium

PCV2 group

PCV2 MOI=1

2.5%-FBS-DMEM medium

PCV2+QUE 12.5 μmol/L

PCV2 MOI=1

12.5 μmol/L QUE

PCV2+QUE 25 μmol/L

PCV2 MOI=1

25 μmol/L QUE

PCV2+QUE 50 μmol/L

PCV2 MOI=1

50 μmol/L QUE

PCV2+QUE 100 μmol/L

PCV2 MOI=1

100 μmol/L QUE

PCV2+QUE 200 μmol/L

PCV2 MOI=1

200 μmol/L QUE

PCV2+QUE 400 μmol/L

PCV2 MOI=1

200 μmol/L QUE

Note: QUE, quercitrin; DMSO: Dimethyl sulfoxide; PCV2, porcine circovirus type 2.

2.3.2. DCFH-DA detection for ROS level in 3D4/2 cells

The experiment was divided into cell control group, PCV2 positive control group, quercitrin treatment groups in three concentrations of (100 μmol/L, 50 μmol/L, 25 μmol/L) and NAC (1 mmol/mL) antioxidant positive control group (As showed in Table 3). The cell concentration was adjusted to 2×105 cells/mL, and cells were plated at 1.0 mL/well in 24-well plates. Afterwards, the cells were incubated at 37°C and 5% CO2 incubator for 12 h and 24 h respectively. Afterwards, the DCFH-DA was diluted with serum-free DMEM culture medium at 1:1000, and 300 μL of DCFH-DA probe was added to each well of the 24-well plate and incubated for 20 min protected from light. The cells were washed 3 times with serum-free DMEM to remove the fluorescent probes that did not entered the cells. Added 1.0 mL of PBS to each well, scraped off the cells with a cell spatula, adjusted the cell concentration to 1×105 cells/mL, and inoculate 200 μL per well into a 96-well black plate to detecting the value of OD, with the excitation wavelength at 488 nm and the emission wavelength at 525 nm.

Table 3 Experimental grouping of ROS level of QUE on PCV2-infected 3D4/2 cells

Groups

Treatment of virus

Treatments

Cell control group

Serum-free medium

2.5%-FBS-DMEM medium

PCV2 group

PCV2 MOI=1.0

2.5%-FBS-DMEM medium

PCV2+QUE 25 μmol/L

PCV2 MOI=1.0  

25 μmol/L QUE

PCV2+QUE 50 μmol/L

PCV2 MOI=1.0  

50 μmol/L QUE

PCV2+QUE 100 μmol/L

PCV2 MOI=1.0  

100 μmol/L QUE

NAC 1 mmol/mL

PCV2 MOI=1.0  

1 mmol/mL NAC

Note: QUE, quercitrin; PCV2, porcine circovirus type 2; NAC, Acetylcysteine.

2.3.3. ELISA detection for the activity of acetylase in 3D4/2 cells

The experiment was divided into cell control group, PCV2 infection group, and QUE groups with three concentrations of 100 μmol/L, 50 μmol/L, 25 μmol/L, with three replicates each group (As showed in Table 4). The cell concentration was adjusted to 5×105 cells/mL, and the cells were inoculated in 12-well plates at 1 mL/well and incubated overnight at 37℃ in 5% CO2. Afterwards, the culture medium was discarded and the cells were washed 3 times with PBS. The PCV2 infection and QUE groups were added with PCV2 virus solution (MOI=1.0), while the cell control group was added with an equal amount of serum-free DMEM medium. And all the plates were incubated for 2 h at 37℃ with 5% CO2, and then the PCV2 solution was aspirated and discarded. After washing the cells with PBS three times, 1.0 mL of 2.5%-FBS-DMEM medium was added to each well for the cell control and PCV2 infection groups. Three concentrations of QUE (100 μmol/L, 50 μmol/L, 25 μmol/L) diluted by 2.5%-FBS-DMEM medium were added to the QUE groups. Cells were incubated at 37°C and 5% CO2 incubator for 8 h, 12 h, 24 h or 36 h. Then the samples were collected and repeatedly freeze-thawed three times. Afterwards, the cell supernatant was collected after centrifugation at 3000 rpm/min for 20 min and stored at -80℃. The enzymatic activities of HAT and HDAC in the cell samples were determined according to the ELISA kit instructions.

Table 4 Experimental grouping of the activity of acetylase in 3D4/2 cells

Groups

Treatment of virus

Treatments

Cell control group

Serum-free medium

2.5%-FBS-DMEM medium

PCV2 group

PCV2 MOI=1.0  

2.5%-FBS-DMEM medium

PCV2+QUE 25 μmol/L

PCV2 MOI=1.0  

25 μmol/L QUE

PCV2+QUE 50 μmol/L

PCV2 MOI=1.0  

50 μmol/L QUE

PCV2+QUE 100 μmol/L

PCV2 MOI=1.0  

100 μmol/L QUE

Note: QUE, quercitrin; PCV2, porcine circovirus type 2.

2.3.4. q-PCR detection for the expression level of mRNA in 3D4/2 cells

The experiment grouping is the same as in 2.3.3. All the cells were incubated for 8 h, 12 h, 24 h or 36 h for subsequent detection. Briefly, total RNA extraction was performed on the collected samples. After RNA reverse transcription to cDNA, the mRNA expression levels of HAT1 and HDAC1 in 3D4/2 cells were detected by q-PCR. And the relative expression of the target genes was calculated using 2-△△ct with β-actin as the reference gene. Gene sequences are shown in the Supplementary Material.

2.3.5. Western blot detection for the expression levels of proteins in 3D4/2

The experiment grouping is the same as in 2.3.3. All the cells were incubated for 24 h for subsequent detection. Specifically, total proteins were first performed on the collected samples. Then, the sample protein concentration was determined according to the BCA kit instructions. Afterwards, the samples were subjected to SDS-PAGE electrophoresis and transferred to PVDF membranes using the constant current method for 1.5 h. Furthermore, the PVDF membranes were closed overnight at 4°C using 5% skim milk. The next day, PVDF membranes were washed three times with 1×TBST. Subsequently, the diluted primary antibody solution was incubated with PVDF membranes at 4°C overnight. Similarly, PVDF membranes were washed three times with 1×TBST. The PVDF membranes were placed in the diluted secondary antibody solution and incubated in an incubator at 37°C for 1 h. Finally, ultrasensitive ECL chemiluminescence color development was performed. The grayscale values of the protein bands were analyzed using Image Quant TL analysis software, and the relative grayscale values of the target proteins were compared using GAPDH/β-actin as the internal reference values.

  1. It is strongly recommended that the figures in the results section be enlarged, and the resolution improved. In some of them, in the printed version, it is impossible to see the subtitles. Legend patterns repeat and some graphics details need to be present.

Response: The resolution of the images has been increased as required.

  1. Discussion is too long and overexplained. Please, consider to unshorten.

Response: As required, we made partial modifications. However, we explored the effect of QUE on PCV2-infected 3D4/2 cells, mainly exploring the relationship among oxidative stress, inflammatory response, and histone acetylation. The discussion section carries out a sufficient discussion of the data in each section of the article in conjunction with previous related studies.

Minor comments:

  1. Page 2, lines 47-59: several experimental models are mixed, and a relation with citrus flavonoids suddenly appear. It is suggested to remove or redo.

Response: As required, we removed the part of “citrus flavonoids”.

  1. Page 2, lines 71-74: please, reference this paragraph.

Response: As required, the reference was added (Anti-inflammatory effects of flavonoids).

  1. Page 2, lines 75-77: please, reference the previously established model.

Response: As required, the reference was added (The effect of Panax notoginseng saponins on oxidative stress induced by PCV2 infection in immune cells: in vitro and in vivo studies).

  1. Page 2, lines 77-81: Aim is not clear.

Response: As required, the aim has been revised as follow: “Based on the previously established model of inflammation and oxidative stress in PCV2-infected 3D4/2 cells, we investigated the regulation of histone acetylation modification of PCV2-infected 3D4/2 cells in vitro by quercitrin and the molecular mechanism of the expression levels of inflammatory and antioxidant factors, aiming to elucidate intervening effects and molecular mechanism of Quercitrin on PCV2-induced histone acetylation, oxidative stress and inflammatory response in 3D4/2 cells, and to provide a theoretical basis for the exploitation of quercitrin and the prevention and treatment of PCVD/PCVAD.

  1. Page 3, lines 103-105: First time that the origin of 3D4/2 cells is mentioned.

Response: This is not the first time we have mentioned 3D4/2 cells. We have already mentioned 3D4/2 cells in the “Abstract” and “Introduction” sections. Here, we just want to explain the origin of 3D4/2 cells in the material section

  1. Page 3, lines 107-123: What is the CCK8 method used for?

Response: As required, the meaning of CCK8 method has been added as followed: “The CCK8 assay was used to detect the effect of quercitrin on the proliferative activity of PCV2-induced 3D4/2 cells in vitro to determine the maximum safe concentration of the drug treating on the cells in vitro. After screening for safe and effective drug concentrations, the regulatory effects of the drug on PCV2-induced histone acetylation, oxidative stress and inflammatory response can be investigated.”

  1. Page 3, line 108: Please clarify why cells are inoculated, and not placed or plated, in culture plates.

Response: The sentence means to seed the cells in 96-well plate, therefore the phrase “inoculated in” from the article has been replaced by “seeded in”.

  1. Page 3, lines 124-132: No referenced method.

Response: We have already changed the subtitle to “DCFH-DA detect the level of ROS in 3D4/2 cells”. And the method was referenced in the “The effect of Panax notoginseng saponins on oxidative stress induced by PCV2 infection in immune cells: in vitro and in vivo studies”.

  1. Page 3, line 132: Which enzyme marker?

Response: DCFH-DA probe was used in the experiment instead of enzyme maker, so the the original sentence has been changed into “Added 1.0 mL of PBS to each well, scraped off the cells with a cell spatula, adjusted the cell concentration to 1×105 cells/mL, and inoculate 200 μL per well into a 96-well black plate to detecting the value of OD, with the excitation wavelength at 488 nm and the emission wavelength at 525 nm.”

  1. Page 3, line 133: How acetylase activity is modulated in the method described in 2.3.3.

Response: We have already revised section 2.3.3, therefore the question about “how the acetylase activity is modulated” has been clarified.

  1. Page 3-4, lines 138-154: Please, specify and reference which gene expression detection method is used here, why the expression of genes related to histone acetylation would reflect their activity, and why in the abstract the method of quantifying HAT and HDAC activity is described as ELISA.

Response: We have already revised section 2.3.3 and 2.3.4, therefore the question about “Page 3-4, lines 138-154” has been clarified.

  1. Page 4, lines 155-162: Authors say that they quantify iNOS, IL-8, NF-kB, HO-1, NQO1, P38, and PI3K/AKT by western blotting. However, in the results section, although WB is really showed, they present IL-6, IL-10, IL-8 and p65 mRNA expression levels. Where are the methods used to quantify these molecules expression? Some of them are even mentioned in Methods.

Response: We used Western blot and q-PCR to quantify those molecules expression. And the specific method has been modified in the material methods section.

  1. Page 4, pages 163-169: Please explain here how replicates were constructed and used.

Response: In our experiments, three biological replicates were designed for each group, and three technical replicates were designed separately for each biological replicate in specific experiments.

  1. Page 4, lines 171-177: Please clarify why you call CCK8 as a proliferative activity assay here. Still, it doesn't seem clear to me which viability decrease you would be dealing with: the one presented by the PCV2 group or the one shown by the 0.05% DMSO group.

Response: The CCK8 method is based on the reduction of WST-8 compounds by certain dehydrogenases in mitochondria to produce orange-colored Formazan dye to detect cytotoxicity and proliferative activity in vitro.

In this study, to better demonstrate the protective effect of quercitrin on the activity of PCV2-infected 3D4/2 cells, and this protective effect was independent of DMSO solvent. First, we measured the cellular viability of 3D4/2 cells treated with quercitrin for 36 h, using the CCK8 method to screen the safe range of the drug for the cells. After that, this range was used as the safe concentration range of the drug for the subsequent experiments of this study. Since quercitrin is insoluble in water, thus, DMSO concentration was diluted to 0.05% using serum-free DMEM for dissolving the drug. The results showed no significant effect of 0.05% DMSO solvent control group on cell proliferation, indicating that quercitrin significantly promoted the proliferative activity of 3D4/2 cells in the effective concentration range, and this promotion was independent of DMSO solvent. Next, we then assayed the viability of proliferation of PCV2-infected 3D4/2 cells in vitro that treated with quercitrin by the CCK8 assay. The findings revealed that PCV2-infected 3D4/2 cells resulted in a decrease in cell viability. The results indicated that PCV2-infected 3D4/2 cells resulted in a decrease in cell viability. In contrast, the additions of 12.5 μmol/L ~ 400 μmol/L quercitrin treated for 36 h promoted the activities of PCV2-infected cells extremely significantly. In addition, the 0.05% DMSO solvent group (PCV2+0.05% DMSO) had no significant effect on the active proliferation of infected cells compared with the PCV2-infected group. These indicated that the decrease of the PCV2-infected 3D4/2 cells viability was not associated with the 0.05% DMSO. The above suggests that quercitrin has a protective effect on the activity of PCV2-infected 3D4/2 cells and is independent of DMSO solvent.

  1. Page 4, lines 177-179: This sentence is interpretative in a Results session.

Response: As required, we have deleted the interpretative sentence: “The results suggested that QUE has a protective effect on the activity of PCV2-infected 3D4/2 cells and was independent of DMSO”.

  1. Page 5, lines 192-193: Why do authors present total ROS with HAT activity? How are they supposably related here?

Response: Viral infection of immune cells induces oxidative stress and inflammatory response, which in turn leads to the accumulation of ROS in the cells. ROS further aggravate oxidative cellular damage by regulating cellular histone acetylation levels and affecting the transcription and expression of immune cell-related genes. So in this paragraph, the results of ROS and histone acetylation modifications are described.

  1. Page 5, lines 209-211: This sentence is interpretative in a Results session.

Response: As required, we have deleted the interpretative sentence.

  1. Page 6, line 233: How ROS levels are regulating histone acetylation here? Please, clarify.

Response: PCV2 induces excessive ROS production in cells, leading to a state of oxidative stress, which in turn leads to an imbalance in the level of cellular acetylation modifications. Therefore, that's how ROS levels regulate histone acetylation levels.

  1. Page 6, figures 2D – 2E: Please clarify if mRNA expression levels do not have a housekeeping normalization.

Response: The relative mRNA levels were calculated using the comparative 2−ΔΔct method and normalized to that of β-actin.

  1. Page 7, figures 3A – 3D: Please clarify if mRNA expression levels do not have a housekeeping normalization.

Response: The relative mRNA levels were calculated using the comparative 2−ΔΔct method and normalized to that of β-actin.

  1. Page 8, figures 4A – 4D: Please clarify if mRNA expression levels do not have a housekeeping normalization.

Response: The relative mRNA levels were calculated using the comparative 2−ΔΔct method and normalized to that of β-actin.

  1. Page 9, figure 5A – 5E: Please clarify if mRNA expression levels do not have a housekeeping normalization.

Response: The relative mRNA levels were calculated using the comparative 2−ΔΔct method and normalized to that of β-actin.

  1. Page 9, figure 5D: It does not seem very clear to me whether the expression of Nrf2 alone would be enough to prove its participation in the pathway, since its action is only effective upon translocation to the nucleus.

Response: The Keap1-Nrf2 signaling pathway is the classical antioxidant signaling pathway. Its common downstream gene products include NQO1, HO-1, GSTs, GCLC, etc. Among them, two classes of quinone oxidoreductases, NQO1 and HO-1, are important downstream regulatory genes of Nrf2. When mild oxidative stress occurs, the organism regulates oxidative stress by activating the Nrf2/HO-1 pathway in order to eliminate the massive accumulation of ROS. The main mechanism is to bind the Nrf2 to target genes, then promoting the production of antioxidant enzymes such as SOD, HO-1, NQO1 and non-enzymatic antioxidants such as GSH and GPx1, thus improving the antioxidant capacity of the body and reducing oxidative damage. In this study, we found that PCV2 infection caused a decrease in mRNA expression such as SOD, GPx1, Keap1, Nrf2, HO-1 and NQO1 in 3D4/2 cells. In contrast, quercitrin treatment was able to promote the mRNA expression of SOD, GPx1, Keap1, Nrf2, HO-1 and NQO1 in infected cells. It indicates that quercitrin can inhibit the oxidative stress response of infected cells through the antioxidant defense system. In addition, this study found that the expression of NQO1 and HO-1 proteins were down-regulated after PCV2 infection of 3D4/2 cells by WB assay. In contrast, the expression levels of NQO1 and HO-1 proteins were up-regulated after quercitrin action on PCV2-infected 3D4/2 cells. It indicates that quercitrin can promote the activation of Nrf2 signaling pathway.

  1. Page 10, Discussion: There are many different routes of secondary AO produced by Nrf2/KEAP. Two important things should be considered here: a) How to differentiate the antioxidant from the anti-inflammatory effect, if they are linked and the results are similar? In addition, The AO effects involved in inflammation involve the Xanthine/XO system. This system was not evaluated and is related to the secondary response in Nrf2/KEAP system. Please, consider that.

Response: We detected the expression of oxidative stress-related indicators such as Gpx1, SOD, HO-1, Keap1, NQO1, Nrf2, and inflammatory response-related indicators such as IL-6, IL-8, IL-10, IκB, etc. Our results suggested that inflammatory response is correlated with oxidative stress, which can induce cellular inflammation, and inflammation can inhibit antioxidant expression systems. The two interact and participate in the process of disease onset and progression. For example, the activation of inflammatory cells such as macrophages leads to an increase in NADPH and iNOS, which in turn induces cells to produce excess ROS. Excess ROS, in turn, causes inflammatory lesions, which in turn elevates oxidative stress levels and exposes the organism to oxidative damage. In addition, excess ROS, after causing oxidative stress in the organism, stimulates intracellular signaling cascades such as NF-κB to enhance the expression of pro-inflammatory factors. (Portugal, M. et al. 2007)

Reviewer 2 Report

The authors investigated the effect of quercitrin on PCV2-induced oxidative stress and inflammatory responses in 3D4/2 cells in vitro. In addition the modulation of histone acetylation modifications were investigated.    Some points:   -Improve the quality of figures 3-5   -Please add a setting with an antioxidant e.g., NAC to validate ROS induction.   

-Dichlorofluorescein diacetate was used to evaluate iintracellular  ROS. The use of insensitive [C-369; 5-(and-6)-carboxy-2¢,7¢-dichlorofluorescein diacetate] fluorescent dyes was not used. The oxidation insensitive probe could be utilized to control for changes in uptake, ester cleavage, and efflux so that differences in fluorescence can definitively be attributed to changes in oxidation of the probe. 

  -What about polyphenol bioavailability in 3D4/2 cells?   -Dichlorofluorescein diacetate was used to evaluate iintracellular  ROS. The use of insensitive [C-369; 5-(and-6)-carboxy-2¢,7¢-dichlorofluorescein diacetate] fluorescent dyes was not used. The oxidation insensitive probe could be utilized to control for changes in uptake, ester cleavage, and efflux so that differences in fluorescence can definitively be attributed to changes in oxidation of the probe.    -Can the levels of quercitrin used in the in vitro assays be attained in animals?   -All polyphenols possess notable reducibility properties. How can the authors rule out this interference into the outcome? That is, the eased oxidative stress could largely attribute to the reducibility properties

Author Response

Reviewer 2:

  1. Improve the quality of figures 3-5.

Response: The quality of the pictures has been modified as required.

  1. Please add a setting with an antioxidant e.g., NAC to validate ROS induction.

Response: As required, we have set the NAC antioxidant group.

3.Dichlorofluorescein diacetate was used to evaluate iintracellular  ROS. The use of insensitive [C-369; 5-(and-6)-carboxy-2¢,7¢-dichlorofluorescein diacetate] fluorescent dyes was not used. The oxidation insensitive probe could be utilized to control for changes in uptake, ester cleavage, and efflux so that differences in fluorescence can definitively be attributed to changes in oxidation of the probe.

Response: We used the fluorescent probe DCFH-DA for the detection of ROS. DCFH-DA itself is not fluorescent and can freely pass through the cell membrane. After it enters the cell, it can be hydrolyzed by intracellular esterases to DCFH, which is not permeable to the cell membrane, thus allowing the probe to be easily loaded into the cell. The intracellular ROS can oxidize the non-fluorescent DCFH to generate fluorescent DCF, and the level of intracellular ROS can be known by detecting the fluorescence of DCF. In addition, referring to the existing related studies on the detection of intracellular ROS, the fluorescent probe DCFH-DA was selected for our experiment to detect the expression level of cellular ROS.

References:

[1] Li JJ, Tang Q, Li Y, et al. Role of oxidative stress in the apoptosis of hepatocellular carcinoma induced by combination of arsenic trioxide and ascorbic acid. Acta Pharmacol Sin. 2006;27(8):1078-1084. doi:10.1111/j.1745-7254. 2006.00345.x

[2] Caro AA, Thompson S, Tackett J. Increased oxidative stress and cytotoxicity by hydrogen sulfide in HepG2 cells overexpressing cytochrome P450 2E1. Cell Biol Toxicol. 2011;27(6):439-453. doi:10.1007/s10565-011-9198-2

[3] Zhang Z, Miao L, Lv C, et al. Wentilactone B induces G2/M phase arrest and apoptosis via the Ras/Raf/MAPK signaling pathway in human hepatoma SMMC-7721 cells. Cell Death Dis. 2013;4(6):e657.. doi:10.1038/cddis.2013.182

[4] Dutta R, Krishnan A, Meng J, et al. Morphine modulation of toll-like receptors in microglial cells potentiates neuropathogenesis in a HIV-1 model of coinfection with pneumococcal pneumoniae. J Neurosci. 2012;32(29):9917-9930. doi:10.1523/JNEUROSCI.0870-12.2012

  1. What about polyphenol bioavailability in 3D4/2 cells?

Response: The study reported that the bioavailability of quercetin was 46.43% (https://old.tcmsp-e.com/molecule.php?qn=98) and that of quercitrin was 4.04% (https://old.tcmsp-e.com/molecule.php?qn= 701). Additionally, quercetin is a hydrolysis product of quercitrin, which is condensed with glucose to form quercitrin.

  1. Can the levels of quercitrin used in the in vitro assays be attained in animals?

Response: We will next further studies in animal experiments. The exact dose in animals is being mapped out, but reports suggest that this formula can be used for in vivo and in vitro drug conversions:

  1. All polyphenols possess notable reducibility properties. How can the authors rule out this interference into the outcome? That is, the eased oxidative stress could largely attribute to the reducibility properties

Response: Que is a polyphenols compound that does have some reducing properties and may have a reducing effect on some extracellular substances with oxidative properties (e.g. O2-). However, we examined the gene expression as well as protein expression of substances related to antioxidant activity (such as SOD, GPX1, Nrf2, HO-1, Keap1, NQO1, etc.) in the cells. An increase in the gene expression and protein expression of these antioxidant activity-related substances in cells indicates that QUE can exert antioxidant effects by regulating cellular signaling pathways.

Reviewer 3 Report

For this manuscript the authors are investigating  the effect of Quercitrin on cellular damage induced by Porcine circovirus type 2 (PCV2), which will eventually result in post-weaning multisystemic Wasting Syndrome (PMWS) and other life-threatening syndromes in baby pigs.

In particular the authors investigated if Quercitrin has can reduce free radicals formation, prevents cell damaging effects induced by free radicals and inflammation.

Using 3D4/2 cells infected with PCV2 (as their model of inflammation), the authors reported that Quercitrin was able to significantly reduce some of the detrimental effects of PVC2 on cellular proliferation and prevent ROS formation.

Moreover, Quercitrin prevented the PCV2-induced increase in histone acyltransferase HAT1 mRNA expression and reduce its activity. Overall, Quercitrin has a beneficial effect in reducing the inflammatory response induced by the virus infection and can prevent the dysregulation of cellular pathways involved in inflammation and ROS regulation.

The authors have produced a considerate amount of data and investigated in details the regulation of inflammation and ROS signaling in relation to PVC2 infection and Quercitrin treatment, and elucidated possible cellular effects of Quercitrin that could translate to in vivo treatment.

However, some corrections and clarifications are needed:

In particular the authors should make sure that the results described in the results section and/or figure legend correspond to the data presented in the figures (see specific examples below).

Introduction:

Page 2, line 57-59: “ In addition, the flavonoid monomeric compound Quercitrin could also promote cartilage extracellular matrix degradation to retard the development of osteoarthritis in ACLT rats [7]”.

This sentence is inexact and the reference cited is incorrect:

  • Quercitrin is reported to alleviate and not promote cartilage extracellular matrix degradation (see Guo et al. 2021 J Adv Res (28) Pages 255-267).
  • Ref# 7 . The results of this manuscript are not relevant here ( ref # 7: Song et al. Saururus chinensis-controlled allergic pulmonary disease through NF-κB/COX-2 and PGE2 pathways. PeerJ. 2020, 8, e10043.

Page 2, li 60-65: Correct sentence too many repetitions.

Page 2: line 72-74:  Quercitrin belongs to the natural flavonoid monomer 72 compounds, which have high exploitation value in the research related to anti-inflammatory and antiviral in animal diseases exploitation. Highlighted part of the sentence is not clear, rephrase.

Results:

Figure 1: for this figure the authors have tested the effect of different concentrations of Quercitrin on PCV2 –induced cell damage. In the figure legend they state that (*) = P < 0.05, so the difference vs. control is significant.  

Results in Figure 1 indicate that DMSO caused a significant decrease in cell proliferation (*)? Could Quercitrin, possible prevent this DMSO-induced damaging effect?   

To make the results more clear, the authors should calculate whether there is a significant difference between (vehicle) DMSO and PCV2? And calculate the effect of Quercitrin on DMSO-induced cell death.

Moreover, it is not necessary to write in every the figure legend that:  ** or ## indicates that there is an extremely significant difference compared to control group (P < 0.01).

 Writing:  ## or/** = p< 0.01 is enough.

Page 4, line 172-177: There is a discrepancy between Figure 1 and the description of the results. The authors stated that “After PCV2 infecting 3D4/2 cells, the cell viability de-172 creased significantly when compared with the cell control group (P < 0.05).” However in the Figure 1 PVC2 vs. Control is (**) = 0.01 ?

The authors also stated that:” 0.05% DMSO had no significant effect on the proliferative activity of PCV2-infected cells”. However, these results are not reported in Figure 1.

Page 5, line 95: “secretory level of ROS”, please rewrite this statement is incorrect. ROS are not secreted but produced. The authors are measuring ROS with the DCFH-DA florescence method so intracellularly? Moreover, ROS are not secreted, but can diffuse across the membrane. 

Page 7, line 248-249: Referring to figure 3 panel C, the authors stated that: “IL-10 mRNA expression level was increased in PCV2 -infected 3D4/2 cells at 8 h, 12 h or24 h post infection (P < 0.05)”. In contrast the graph in figure 3 panel C show a decrease in the expression of IL-10 mRNA after PVC2 alone.

Page 10, figure 6. The blots are overexposed. When the signal is saturated it is not possible to do appropriate data analysis. The authors should show a better example

Author Response

Reviewer 3

1.Page 2, line 57-59: “ In addition, the flavonoid monomeric compound Quercitrin could also promote cartilage extracellular matrix degradation to retard the development of osteoarthritis in ACLT rats [7]”.This sentence is inexact and the reference cited is incorrect:Quercitrin is reported to alleviate and not promote cartilage extracellular matrix degradation (see Guo et al. 2021 J Adv Res (28) Pages 255-267).

Ref# 7 . The results of this manuscript are not relevant here ( ref # 7: Song et al. Saururus chinensis-controlled allergic pulmonary disease through NF-κB/COX-2 and PGE2 pathways. PeerJ. 2020, 8, e10043.

Response:: As required, we have revised the sentence as: In addition, the flavonoid monomeric compound quercitrin could also recover pulmonary epithelial cell hyperplasia and inflammatory cell infiltrationimprove via NF-κB/COX-2 and PGE2 signaling pathway [10].

  1. Page 2, li 60-65: Correct sentence too many repetitions.Page 2: line 72-74: Quercitrin belongs to the natural flavonoid monomer 72 compounds, which have high exploitation value in the research related to anti-inflammatory and antiviral in animal diseases exploitation. Highlighted part of the sentence is not clear, rephrase.

Response: As required we have rephrased the sentence as: Quercitrin belongs to the natural flavonoid monomer compounds and thus has the same biological activity. In addition, the development of animal diseases is often associated with inflammation or viral infections. Therefore, it is worthwhile to develop and exploit quercitrin for its role in the field of animal diseases.

  1. Results:

Figure 1: for this figure the authors have tested the effect of different concentrations of Quercitrin on PCV2 –induced cell damage. In the figure legend they state that (*) = P < 0.05, so the difference vs. control is significant.

Results in Figure 1 indicate that DMSO caused a significant decrease in cell proliferation (*)? Could Quercitrin, possible prevent this DMSO-induced damaging effect?

To make the results more clear, the authors should calculate whether there is a significant difference between (vehicle) DMSO and PCV2? And calculate the effect of Quercitrin on DMSO-induced cell death.

Moreover, it is not necessary to write in every the figure legend that: ** or ## indicates that there is an extremely significant difference compared to control group (P < 0.01).

Writing: ## or/** = p< 0.01 is enough.

Response: In this study, to better demonstrate the protective effect of quercitrin on the activity of PCV2-infected 3D4/2 cells, and this protective effect was independent of DMSO solvent. First, we measured the cellular viability of 3D4/2 cells treated with quercitrin for 36 h, using the CCK8 method to screen the safe range of the drug for the cells. After that, used this range as the safe concentration range of the drug used in the subsequent experiments of this study. Since quercitrin is insoluble in water, thus, DMSO concentration was diluted to 0.05% using serum-free DMEM for dissolving the drug. The results showed no significant effect of 0.05% DMSO solvent control group on cell proliferation, indicating that quercitrin significantly promoted the proliferative activity of 3D4/2 cells in the effective concentration range, and this promotion was independent of DMSO solvent. Next, we then assayed the viability of proliferation of PCV2-infected 3D4/2 cells in vitro that treated with quercitrin by the CCK8 assay. The results indicated that PCV2-infected 3D4/2 cells resulted in a decrease in cell viability. The results indicated that PCV2-infected 3D4/2 cells resulted in a decrease in cell viability. In contrast, the addition of 12.5 μmol/L ~ 400 μmol/L quercitrin treated for 36 h all promoted the activity of PCV2-infected cells extremely significantly. In addition, the 0.05% DMSO solvent group (PCV2+0.05% DMSO) had no significant effect on the active proliferation of infected cells compared with the PCV2-infected group. These indicate that the decrease of the PCV2-infected 3D4/2 cells viability was not associated with the 0.05% DMSO. The above suggests that quercitrin has a protective effect on the activity of PCV2-infected 3D4/2 cells and is independent of DMSO solvent.

  1. Page 4, line 172-177: There is a discrepancy between Figure 1 and the description of the results. The authors stated that “After PCV2 infecting 3D4/2 cells, the cell viability de-172 creased significantly when compared with the cell control group (P < 0.05).” However in the Figure 1 PVC2 vs. Control is (**) = 0.01 ? The authors also stated that:” 0.05% DMSO had no significant effect on the proliferative activity of PCV2-infected cells”. However, these results are not reported in Figure 1.

Response: As required, we have revised the results of 3.1 as: As displayed in Figure 1A, QUE at 25 ~400 μmol/L significantly promote the activity of 3D4/2 cells, and this promotion was independent of DMSO solvent. As showed in Figure 1B. After PCV2 infecting 3D4/2 cells, the cell viability decreased significantly when compared with the cell control group (P < 0.01). The addition of QUE at 12.5 μmol/L, 25 μmol/L, 50 μmol/L, 100 μmol/L, 200 μmol/L or 400 μmol/L promoted the the activity of PCV2-infected cells (P < 0.01). In addition, the activity of PCV2-infected 3D4/2 cells elevated with QUE treatment. Besides 0.05% DMSO had no significant effect on the activity of PCV2-infected cells.

  1. Page 5, line 95: “secretory level of ROS”, please rewrite this statement is incorrect. ROS are not secreted but produced. The authors are measuring ROS with the DCFH-DA florescence method so intracellularly? Moreover, ROS are not secreted, but can diffuse across the membrane.

Response: As required, we have changed the word “secretory” into “production”. Besides, the ROS were detected with 2.7-dichlorodihydrofluorescein diacetate (DCFH-DA).

  1. Page 7, line 248-249: Referring to figure 3 panel C, the authors stated that: “IL-10 mRNA expression level was increased in PCV2 -infected 3D4/2 cells at 8 h, 12 h or24 h post infection (P < 0.05)”. In contrast the graph in figure 3 panel C show a decrease in the expression of IL-10 mRNA after PVC2 alone.

Response: As required, we have revised the results of 3.3 as: As shown in Figure 3, PCV2 infection in 3D4/2 cells was able to up-regulate the expression levels of cellular IL-6 and IL-8 mRNA at 8 h, 12 h, 24 h or 36 h when compared with cell control group (P < 0.05 or P < 0.01). In addition, p65 mRNA expression level was decreased in PCV2-infected 3D4/2 cells at 8 h, 12 h or 24 h post infection (P < 0.05, or P < 0.01) and IL-10 mRNA expression level was decreased in PCV2 -infected 3D4/2 cells at 8 h, 12 h post infection (P < 0.05) . Compared with the PCV2 infection group, QUE from 25 to 100 μmol/L significantly down-regulated the mRNA expression levels of IL-6 and IL-8 at 8 h or 36 h post treatment (P < 0.05 or P < 0.01) (see also Fig. 3A, 3B) and significantly up-regulated mRNA expression of p65 at 12 h and 24 h post treatment (P < 0.01 or P < 0.05) (Figure 3C). QUE at 100 μmol/L significantly up-regulated p65 mRNA gene expression at 8 h, 12 h or 24 h post treatment (P < 0.05) (Figure 3C). QUE at 25 μmol/L, 50 μmol/L or 100 μmol/L extremely up-regulated mRNA expression level of IL-10 (P < 0.01) at 12 h or 24 h post treatment. Besides, 50 μmol/L and 100 μmol/L of QUE extremely up-regulated IL-10 mRNA expression at 8 h and 36 h post treatment (P < 0.01). (Figure 3D).

  1. Page 10, figure 6. The blots are overexposed. When the signal is saturated it is not possible to do appropriate data analysis. The authors should show a better example

Response: We have provided the original images of the protein bands. We do not think they are overexposed. It is possible that the low resolution of the previous figures made you think that our blots were overexposed. We have now re-uploaded the high-resolution figures.

Reviewer 4 Report

The article “Intervening effects and molecular mechanismof Quercitrin on PCV2-induced histone acetylation, oxidative stress and inflammatory response in 3D4/2 cells” by Qi Chen, Yuheng Wei, Yi Zhao, Xiaodong Xie, Na Kuang, Yingyi Wei, Meiling Yu and Tingjun Hu corresponds to the theme of the journal Antioxidants. However, it requires significant revision before it is accepted for printing.

Key remarks:

  1. Figure 1:  a) Explain the formation of a separate group "0.05% DMSO". The substance was dissolved in DMSO? Why was it impossible to make a group of intact cells + DMSO?   Based on your graph, DMSO reduces the survival of intact cells, how can you explain this?  b) Not sure about thesignificantceof the data, based on the huge error, especially the group with quercitrin 12.5 μM. And also between groups "Control" and "0.05% DMSO"  c) in the caption to the figure “Shoulder mark # indicates significant difference compared with PCV2 group (P < 0.05)”, but there is no such designation on the graph.
  2. Why, after the experiment on cell viability, concentrations of 25, 50 and 100 µM were taken for further research, if the survival rate is good at 400 µM?
  3. Figure 2 - 5 is worth changing, low resolution. Is it possible to present some of the results in the form of a table with primary data? It is also worth making the same color captions for the graphs (do both controls in the same way, in Figure 2 the PCV2 group is blue, and then red; in addition, the concentration of the test compound in Figure 2 is from 100 μM to 25, and then vice versa)
  4. Line 196-198 does not indicate in the text that there are differences in the groups of PCV2+ substance from the intact control, although * is on the graph (there are such inconsistencies in the description of other figures)
  5. Correction of ANOVA is not indicated in the description of statistics.
  6. In all subparagraphs of the “results” chapter, the description of the graphs is difficult to understand, since there is a constant jump from one data to another (for example, they write about 100 μM, then about other concentrations, then again about 100 μM).

203-204 I propose to systematize the description either by targets, or by concentration, or by time.

  1. Description for Figure 2:  a) Lines 203-204 "Compared with the PCV2 infection group, QUE at 50 μmol/L significantly decreased the HAT activity at 8 h, 12 h or 24 h post treatment (P < 204 0.05).", and why the decrease is not indicated and at 36 h, is the confidence value indicated on the graph? (and there are many such non-responses in the results section)  b) Lines 208-209 "QUE at 25~100 μmol/L significantly decreased the HAT and HDAC ac-208 tivities at 36 h post treatment (P < 0.01 or P < 0.05)". But in graph 2c, QUE in µM at 36 h is not significant, so not all concentrations reduce HDAC activity?  c) Lines 206-207 "QUE at 50 μmol/L and 100 μmol/L significantly decreased HAT activity and increased HDAC activity (P < 0.01) at 24 h post treatment". It is not indicated what was compared against, and also on the graph the values ​​P < 0.01 or P < 0.05, in the text only (P < 0.01)  d) Line 212-214 "As shown in Figure 2D, 2E. Compared with cell control group, PCV2 infection significantly up-regulated the expression levelof HAT1 mRNA and significantly down-regulated the expression level of HDAC1 mRNA in 3D4/2 cells at 8 h, 12 h, 24 hor 36 h post infection (P < 0.05 or P < 0.01)”. No reliability to HDAC at 36 hours  e) Figure 2f - make the same color codes with the previous charts. In the AcH3 graph, the control and PCV2 are unlikely to be reliable, the errors overlap each other, in AcH4 - for 25 and 100 as well.
  2. Description Figure 3  a) Figure 3a 12 hours two controls are not significant. Make the color and sequence of concentrations the same with the previous pictures.  b) Lines 247-249 "In addition, p65 mRNA expression level was decreased and IL-10 mRNA expression level was increased in PCV2 -infected 3D4/2 cells at 8 h, 12 h or24 h post infection(P < 0.05)" in graph 3d, the expression level of IL-10 mRNA decreased after 8 and 12 hours, text and graphics do not match

And many more inconsistencies graph-text

  1. Description Figure 4:  a) Figure 4e - Line 273-276 “Compared with the cell control group, PCV2-infected 3D4/2 cells at 8 h, 12 h or 24 h was able to significantly up-regulate intracellular mRNA expression levels of IκB and AKT and down -regulate intracellular mRNA expression levels of SOD and GPx1." There is no drop in the graph at 8 hours between these groups. And after 24 hours, Figure 4c is not sure that there is significant, the errors are large.
  2. Figure 6 it is necessary to indicate that the time period is 24 hours (indicated in the text).Remove unnecessary designations **' and ## , they are not on the charts. There are doubts that there are significant differences in control and PCV2 (Figure 6e)
  3. Figure 7 remove unnecessary designations **' and # from the description; indicate that the time period is 24 hours (indicated in the text).; increase in IκB (difference between controls, also in doubt);
  4. Figure 8 remove unnecessary designations ##
  5. Conclusion: Lines 487-489 “After treatment of PCV2-infected cells with QUE, three concentrations of QUE provided effective control of the cellular inflammatory response by effectively inhibiting the mRNA expression levels of the inflammatory factors IL-6, IL-8, and IL-10", however, in Figure 3e with IL-10, all compounds increase the expression of IL-10 mRNA!

Author Response

Reviewer 4:

  1. Figure 1: a) Explain the formation of a separate group "0.05% DMSO". The substance was dissolved in DMSO? Why was it impossible to make a group of intact cells + DMSO? Based on your graph, DMSO reduces the survival of intact cells, how can you explain this? b) Not sure about the significant of the data, based on the huge error, especially the group with quercitrin 12.5 μM. And also between groups "Control" and "0.05% DMSO" c) in the caption to the figure “Shoulder mark # indicates significant difference compared with PCV2 group (P < 0.05)”, but there is no such designation on the graph.

Response: a) In this study, to better demonstrate the protective effect of quercitrin on the activity of PCV2-infected 3D4/2 cells, and this protective effect was independent of DMSO solvent. Firstly, we measured the cellular viability of 3D4/2 cells treated with quercitrin for 36 h, using the CCK8 method to screen the safe range of the drug for the cells. After that, used this range as the safe concentration range of the drug used in the subsequent experiments of this study. Since quercitrin is insoluble in water, thus, DMSO concentration was diluted to 0.05% using serum-free DMEM for dissolving the drug. The results showed no significant effect of 0.05% DMSO solvent control group on cell proliferation, indicating that quercitrin significantly promoted the proliferative activity of 3D4/2 cells in the effective concentration range, and this promotion was independent of DMSO solvent. Next, we then assayed the viability of proliferation of PCV2-infected 3D4/2 cells in vitro that treated with quercitrin by the CCK8 assay. The results indicated that PCV2-infected 3D4/2 cells resulted in a decrease in cell viability. The results indicated that PCV2-infected 3D4/2 cells resulted in a decrease in cell viability. In contrast, the addition of 12.5 μmol/L ~ 400 μmol/L quercitrin treated for 36 h all promoted the activity of PCV2-infected cells extremely significantly. In addition, the 0.05% DMSO solvent group (PCV2+0.05% DMSO) had no significant effect on the active proliferation of infected cells compared with the PCV2-infected group. These indicate that the decrease of the PCV2-infected 3D4/2 cells viability was not associated with the 0.05% DMSO. The above suggests that quercitrin has a protective effect on the activity of PCV2-infected 3D4/2 cells and is independent of DMSO solvent.

  1. b) One-Way ANOVA was used for the experimental data analysis and even though the 12.5 μM QUE group had a larger value of variance, it still showed significant differences from the PCV2 group. Similarly, the data between the blank control group and the 0.05% DMEM group also showed significant differences.
  2. c) The sentence “shoulder mark # indicates significant difference compared with PCV2 group (P < 0.05)” has been deleted.
  3. Why, after the experiment on cell viability, concentrations of 25, 50 and 100 µM were taken for further research, if the survival rate is good at 400 µM?

Response: Our results showed that quercitrin was not toxic to 3D4/2 cells in the  concentrations ranging from 25 μmol/L to 400 μmol/L. PCV2 infection of 3D4/2 cells decreased the activity of the cell. And 25 μmol/L to 100 μmol/L of quercitrin increased the activity of PCV2-infected 3D4/2 cells with increasing drug concentrations. The drug concentrations of 25 μmol/L, 50 μmol/L, and 100 μmol/L were selected as the drug concentrations used in the subsequent experiments, taking into account the drug cost and effectiveness.

  1. Figure 2 - 5 is worth changing, low resolution. Is it possible to present some of the results in the form of a table with primary data? It is also worth making the same color captions for the graphs (do both controls in the same way, in Figure 2 the PCV2 group is blue, and then red; in addition, the concentration of the test compound in Figure 2 is from 100 μM to 25, and then vice versa).

Response: The Figure has been modified as required.

  1. Line 196-198 does not indicate in the text that there are differences in the groups of PCV2+ substance from the intact control, although * is on the graph (there are such inconsistencies in the description of other figures).

Response: Modified as required.

  1. Correction of ANOVA is not indicated in the description of statistics.

Response: We use the Dunnet’s test to correct the ANOVA, and we have changed the description of statistics into: “Statistical analyses were performed using one-way analysis of variance (One-Way ANOVA) followed by Dunnet’s Test and were analyzed with SPSS 23.0 software (IBM Corp., USA). The data are shown as the mean ± SD. The shoulder labels * or ** indicate significant or extremely significant differences (P < 0.05 or P < 0.01) respectively for each test group compared with the cellular control group; the shoulder labels # or ## indicate significant or extremely significant differences (P < 0.05 or P < 0.01) respectively for each test group compared with the PCV2-infected group.

  1. In all subparagraphs of the “results” chapter, the description of the graphs is difficult to understand, since there is a constant jump from one data to another (for example, they write about 100 μM, then about other concentrations, then again about 100 μM).

203-204 I propose to systematize the description either by targets, or by concentration, or by time.

Response: We have revised as required

  1. Description for Figure 2: a) Lines 203-204 "Compared with the PCV2 infection group, QUE at 50 μmol/L significantly decreased the HAT activity at 8 h, 12 h or 24 h post treatment (P < 204 0.05).", and why the decrease is not indicated and at 36 h, is the confidence value indicated on the graph? (and there are many such non-responses in the results section) b) Lines 208-209 "QUE at 25~100 μmol/L significantly decreased the HAT and HDAC ac-208 tivities at 36 h post treatment (P < 0.01 or P < 0.05)". But in graph 2c, QUE in µM at 36 h is not significant, so not all concentrations reduce HDAC activity? c) Lines 206-207 "QUE at 50 μmol/L and 100 μmol/L significantly decreased HAT activity and increased HDAC activity (P < 0.01) at 24 h post treatment". It is not indicated what was compared against, and also on the graph the values P < 0.01 or P < 0.05, in the text only (P < 0.01) d) Line 212-214 "As shown in Figure 2D, 2E. Compared with cell control group, PCV2 infection significantly up-regulated the expression level of HAT1 mRNA and significantly down-regulated the expression level of HDAC1 mRNA in 3D4/2 cells at 8 h, 12 h, 24 h or 36 h post infection (P < 0.05 or P < 0.01)”. No reliability to HDAC at 36 hourse) Figure 2f - make the same color codes with the previous charts. In the AcH3 graph, the control and PCV2 are unlikely to be reliable, the errors overlap each other, in AcH4 - for 25 and 100 as well.

Response: We have revised as required

  1. Description Figure 3 a) Figure 3a 12 hours two controls are not significant. Make the color and sequence of concentrations the same with the previous pictures. b) Lines 247-249 "In addition, p65 mRNA expression level was decreased and IL-10 mRNA expression level was increased in PCV2 -infected 3D4/2 cells at 8 h, 12 h or 24 h post infection (P < 0.05)" in graph 3 d, the expression level of IL-10 mRNA decreased after 8 and 12 hours, text and graphics do not match.

Response: We have revised as required

  1. Description Figure 4: a) Figure 4e - Line 273-276 “Compared with the cell control group, PCV2-infected 3D4/2 cells at 8 h, 12 h or 24 h was able to significantly up-regulate intracellular mRNA expression levels of IκB and AKT and down -regulate intracellular mRNA expression levels of SOD and GPx1." There is no drop in the graph at 8 hours between these groups. And after 24 hours, Figure 4c is not sure that there is significant, the errors are large.

Response: We have revised as required

  1. Figure 6 it is necessary to indicate that the time period is 24 hours (indicated in the text).Remove unnecessary designations **' and ## , they are not on the charts. There are doubts that there are significant differences in control and PCV2 (Figure 6e).

Response: We have revised as required

  1. Figure 7 remove unnecessary designations **' and # from the description; indicate that the time period is 24 hours (indicated in the text); increase in IκB (difference between controls, also in doubt);

Response: We have revised as required

  1. igure 8 remove unnecessary designations ##.

Response: We have revised as required

  1. Conclusion: Lines 487-489 “After treatment of PCV2-infected cells with QUE, three concentrations of QUE provided effective control of the cellular inflammatory response by effectively inhibiting the mRNA expression levels of the inflammatory factors IL-6, IL-8, and IL-10", however, in Figure 3e with IL-10, all compounds increase the expression of IL-10 mRNA!

Response: We have revised as required

Round 2

Reviewer 3 Report

The authors addressed all the critical concerns raised be the reviewers and significantly improved the manuscript.

One minor point for me, is repetitive use of this sentence throughout the manuscript:

Shoulder mark ‘*’ indicates that there is a significant 240 difference compared with the cell control group (P < 0.05) and ‘**’ indicates that there is an extremely significant difference compared with the cell control group (P < 0.01); Shoulder mark # indicates significant difference compared with PCV2 group (P < 0.05), and shoulder mark ## indicates extremely significant difference compared with PCV2 group (P < 0.01).

I think that this extensive explanation of the symbols relative to the statistic should be limited to the method or results section.

In the figure legend writing: * p<0.5 and ** p<0.01 and # p<0.5 and ## p<0.01 is sufficient .

More over stating that ## or ** indicates that there is an extremely significant difference is inaccurate. The author should stick to general statistic guidelines (see below).

P value

Wording

Summary

< 0.0001

Extremely significant

****

0.0001 to 0.001

Extremely significant

***

0.001 to 0.01

Very significant

**

0.01 to 0.05

Significant

*

≥ 0.05

Not significant

ns

If you choose APA or NEJM formatting for P values, Prism uses this scheme (note the absence of ****).

P value

Wording

Summary

< 0.001

Very significant

***

0.001 to 0.01

Very significant

**

0.01 to 0.05

Significant

*

≥ 0.05

Not significant

ns

Author Response

Reviewer #3:

The authors addressed all the critical concerns raised be the reviewers and significantly improved the manuscript.

One minor point for me, is repetitive use of this sentence throughout the manuscript:

Shoulder mark ‘*’ indicates that there is a significant 240 difference compared with the cell control group (P < 0.05) and ‘**’ indicates that there is an extremely significant difference compared with the cell control group (P < 0.01); Shoulder mark # indicates significant difference compared with PCV2 group (P < 0.05), and shoulder mark ## indicates extremely significant difference compared with PCV2 group (P < 0.01).

I think that this extensive explanation of the symbols relative to the statistic should be limited to the method or results section.

In the figure legend writing: * p<0.5 and ** p<0.01 and # p<0.5 and ## p<0.01 is sufficient .

More over stating that ## or ** indicates that there is an extremely significant difference is inaccurate. The author should stick to general statistic guidelines (see below).

Response: As required, firstly, we have revised content in the part of Data processing and analysis as: One-Way ANOVA (One-Way Analysis of Variance) in SPSS statistics 23 data analysis software was used to analyze the test data, which were expressed by Mean±SD. The shoulder label * indicates significant difference compared with the cell control group (P < 0.05) and ** indicates very significant difference compared with the cell control group (P < 0.01). Besides, the shoulder label # indicates significant difference compared with the PCV2 infection group (P < 0.05) and ## indicates very significant difference compared with the PCV2 infection group (P < 0.01).

Secondly, we have revised “extremely significant” as “very significant”

Thirdly, we have revised the figure legend writing as: * p<0.05 and ** p<0.01 and # p<0.05 and ## p<0.01.

Reviewer 4 Report

I thank the authors for their responses to comments. After this revision the article can be published.

Author Response

Reviewer #4:

I thank the authors for their responses to comments. After this revision the article can be published.

Response: As required, we have corrected the grammar mistakes and checked the spell of the phrases in the revised manuscript.
